# The Nucleosome Remodelling and Deacetylation complex suppresses transcriptional noise during lineage commitment

Thomas Burgold[1,†], Michael Barber[1], Susan Kloet[2,‡], Julie Cramard[1], Sarah Gharbi[1], Robin Floyd[1], Masaki Kinoshita[1], Meryem Ralser[1], Michiel Vermeulen[2] ID, Nicola Reynolds[1], Sabine Dietmann[1] & Brian Hendrich[1,3,*] ID

## Abstract

Multiprotein chromatin remodelling complexes show remarkable conservation of function amongst metazoans, even though components present in invertebrates are often found as multiple paralogous proteins in vertebrate complexes. In some cases, these paralogues specify distinct biochemical and/or functional activities in vertebrate cells. Here, we set out to define the biochemical and functional diversity encoded by one such group of proteins within the mammalian Nucleosome Remodelling and Deacetylation (NuRD) complex: Mta1, Mta2 and Mta3. We find that, in contrast to what has been described in somatic cells, MTA proteins are not mutually exclusive within embryonic stem (ES) cell NuRD and, despite subtle differences in chromatin binding and biochemical interactions, serve largely redundant functions. ES cells lacking all three MTA proteins exhibit complete NuRD loss of function and are viable, allowing us to identify a previously unreported function for NuRD in reducing transcriptional noise, which is essential for maintaining a proper differentiation trajectory during early stages of lineage commitment.

**Keywords** chromatin; ES Cell; lineage commitment; NuRD; transcription
**Subject Categories** Development & Differentiation; Stem Cells; Transcription
**The EMBO Journal (2019) 38: e100788**

## Introduction

Mammalian cells contain a number of proteins capable of using ATP hydrolysis to shift nucleosomes relative to the DNA sequence, thereby facilitating chromatin remodelling. In mammals, these ATP-dependent chromatin remodelling proteins usually exist within multiprotein complexes and play essential roles in the control of gene expression, DNA replication and repair (Hargreaves & Crabtree, 2011; Narlikar et al, 2013; Hota & Bruneau, 2016).

Nucleosome Remodelling and Deacetylation (NuRD) is one such multiprotein complex which is unique in that it contains both chromatin remodelling and protein deacetylase activity. NuRD is highly conserved amongst metazoans and has been shown to play important roles in cell fate decisions in a wide array of systems (Denslow & Wade, 2007; Signolet & Hendrich, 2015). For example, in embryonic stem (ES) cells, NuRD controls nucleosome positioning at regulatory sequences to finely tune gene expression (Reynolds et al, 2012; Bornelöv et al, 2018), and in somatic lineages, NuRD activity has been shown to prevent inappropriate expression of lineage-specific genes to ensure fidelity of somatic lineage decisions (Denner & Rauchman, 2013; Knock et al, 2015; Gomez-Del Arco et al, 2016; Loughran et al, 2017). It was recently demonstrated that this is achieved in both ES cells and B-cell progenitors by restricting access of transcription factors to regulatory sequences (Liang et al, 2017; Loughran et al, 2017; Bornelöv et al, 2018). Additionally, aberrations in expression levels of NuRD component proteins are increasingly being linked to cancer progression (Lai & Wade, 2011; Mohd-Sarip et al, 2017).

Nucleosome Remodelling and Deacetylation is comprised of two enzymatically and biochemically distinct subcomplexes: a chromatin remodelling and a deacetylase subcomplex. The chromatin remodelling subcomplex contains a nucleosome remodelling ATPase protein (Chd3/4/5) along with one of the zinc finger proteins Gatad2a/b and the Doc1/Cdk2ap1 protein, while the deacetylase subcomplex contains class I histone deacetylase proteins Hdac1/2, the histone chaperones Rbbp4/7, the Metastasis Tumour Antigen family of proteins, Mta1, Mta2 and Mta3 and, in pluripotent cells, the zinc finger proteins Sall1/4 (Lauberth & Rauchman, 2006; Allen et al, 2013; Kloet et al, 2015; Bode et al, 2016; Low et al, 2016; Miller et al, 2016; Spruijt et al, 2016; Zhang et al,

1 Wellcome– MRC Stem Cell Institute, University of Cambridge, Cambridge, UK
2 Department of Molecular Biology, Faculty of Science, Radboud Institute for Molecular Life Sciences, Oncode Institute, Radboud University, Nijmegen, The Netherlands
3 Department of Biochemistry, University of Cambridge, Cambridge, UK
*Corresponding author. Tel: +44 1223 760205; E-mail: brian.hendrich@cscr.cam.ac.uk
†Present address: Wellcome Sanger Institute, Cambridge, UK
‡Present address: Leiden Genome Technology Center, Department of Human Genetics, Leiden University Medical Center, Leiden, The Netherlands

2016). These two subcomplexes are bridged by Mbd2/3, creating intact NuRD (Fig 1A). While HDAC and RBBP proteins are also associated with other chromatin-modifying complexes, the MBD, GATAD2 and MTA proteins are obligate NuRD components. Functional and genetic data indicate that the CHD4-containing remodelling subunit is capable of functioning independently of intact NuRD (O'Shaughnessy & Hendrich, 2013; O'Shaughnessy-Kirwan et al, 2015; Ostapcuk et al, 2018). The deacetylase subcomplex has been shown biochemically to exist outside of intact NuRD, though whether this subcomplex has any specific function is not clear (Link et al, 2018; Zhang et al, 2018).

Changes in subunit composition in large multiprotein, chromatin-modifying complexes such as PRC1 and BAF have been shown to correlate with distinct changes in function at sites of action in a cell-type-specific manner (Ho & Crabtree, 2010; Morey et al, 2012). The NuRD complex might therefore be expected to show similar diversity in both composition and function, and in fact, diversification of NuRD function has been described through differential incorporation of various isoforms of NuRD component proteins (Bowen et al, 2004). For example, Mbd2 and Mbd3 are mutually exclusive within NuRD (Le Guezennec et al, 2006). While Mbd2 is not required for mammalian development, Mbd3 is essential at early postimplantation stages in mice (Hendrich et al, 2001). Mbd2/NuRD is a methyl-CpG binding co-repressor complex which is dispensable for early development but Mbd3/NuRD, a transcriptional modulator found at sites of active transcription, has been shown to play important roles in regulation of cell fate decisions in multiple developmental systems (Feng & Zhang, 2001; Reynolds et al, 2012, 2013; Gunther et al, 2013; Shimbo et al, 2013; Menafra et al, 2014). NuRD complexes containing either CHD3, CHD4 or CHD5 play distinct roles during cortical development (Nitarska et al, 2016).

Further functional and biochemical diversification occurs through alternate use of the three MTA proteins within NuRD. MTA proteins function as a scaffold around which the deacetylase subcomplex is formed, comprising a 2:2:4 stoichiometry of MTAs: HDACs:RBBPs (Millard et al, 2013, 2016; Smits et al, 2013; Zhang et al, 2016). While invertebrates predominantly have a single MTA protein orthologue, most vertebrates have three. The three mammalian MTA proteins are highly conserved, differing from each other predominantly at their C-termini (Fig 1A). The MTA1 protein was originally identified because of its elevated expression in metastatic cell lines (Toh et al, 1994), and subsequently, all three MTA proteins have been shown to be upregulated in a range of different cancer types (Covington & Fuqua, 2014; Sen et al, 2014; Ma et al, 2016). The MTA1 and MTA3 proteins were shown to form distinct NuRD complexes in breast cancer cells and in B cells and were recruited by different transcription factors to regulate gene expression (Mazumdar et al, 2001; Fujita et al, 2003, 2004; Si et al, 2015). These studies did not detect biochemical interactions between MTA3 and the other MTA proteins, leading to the conclusion that MTA proteins are mutually exclusive within NuRD. In contrast, Mta1 was shown to interact with Mta2 in MEL cells, indicating that mutual exclusivity may be cell-type-specific (Hong et al, 2005). While all three *Mta* genes are expressed in ES cells, detailed biochemical analysis of interactions of MTA proteins with one another or with the various NuRD components in ES cells has not previously been described.

Functional evidence does not support a strict lack of redundancy amongst MTA proteins during mammalian development. While zygotic deletion of *Chd4* or *Mbd3* results in pre- or peri-implantation developmental failure, respectively (Kaji et al, 2007; O'Shaughnessy-Kirwan et al, 2015), mice deficient in any one of the three MTA proteins show minimal phenotypes. Mice lacking either *Mta1* or *Mta3* are viable and fertile (Manavathi et al, 2007; Mouse Genome Informatics), while mice lacking *Mta2* show incompletely penetrant embryonic lethality and immune system defects (Lu et al, 2008). In the current study, we took a systematic approach to dissecting MTA protein biochemical and functional diversity. We find that, in contrast to what has been described in somatic cells, MTA proteins are not mutually exclusive within NuRD in ES cells and serve largely redundant functions. Furthermore, ES cells lacking all three MTA proteins are viable and represent a complete NuRD null, allowing us to identify a previously undetected function for NuRD in early stages of lineage commitment.

## Results

### MTA proteins are not mutually exclusive within the NuRD complex in ES cells

The absence of a detected interaction between the MTA2 and MTA3 proteins in human cells (Fujita et al, 2003; Si et al, 2015), and the observation that different MTA proteins can show different protein–protein interactions in B cells (Fujita et al, 2004) have led to the conclusion that the MTA proteins are mutually exclusive within NuRD and could hence confer functional diversity to the NuRD complex (Lai & Wade, 2011). To investigate the biochemical specificity of the MTA proteins in an unbiased manner, we used gene targeting of endogenous loci to produce three different mouse ES cell lines in which an epitope tag was fused to the C-terminus of each MTA protein (Fig 1A). Although *MTA* genes show different expression patterns in preimplantation mouse development, all three are expressed in peri-implantation and early postimplantation epiblast, the tissue most similar to the naïve ES cell state (Fig EV1). We therefore considered ES cells to be a good system in which to investigate the function of MTA proteins.

Each tagged protein was expressed at levels comparable to those of wild-type proteins and was found to interact with other NuRD component proteins by immunoprecipitation (Fig 1B). Each MTA protein was also able to immunoprecipitate the other MTA proteins in addition to unmodified forms of itself (Fig 1B). Mta3 was barely detectable in Mta1-3xFLAG immunoprecipitates by IP-western blot (though robustly found by mass spectrometry, see below), but Mta1 was robustly detected associating with Mta3-3xFLAG (Fig 1B). This could indicate that any Mta1-Mta3-containing NuRD complexes represent a relatively small proportion of nuclear Mta1, but a relatively large proportion of nuclear Mta3. Individual NuRD complexes contain two MTA proteins (Millard et al, 2013; Smits et al, 2013; Zhang et al, 2016), so the identification of an interaction between MTA proteins means that NuRD complexes in ES cells could contain either homodimers or heterodimers consisting of any combination of the three MTA proteins.

To investigate the potential biochemical diversification conferred by different MTA proteins in pluripotent cells, we used our tagged

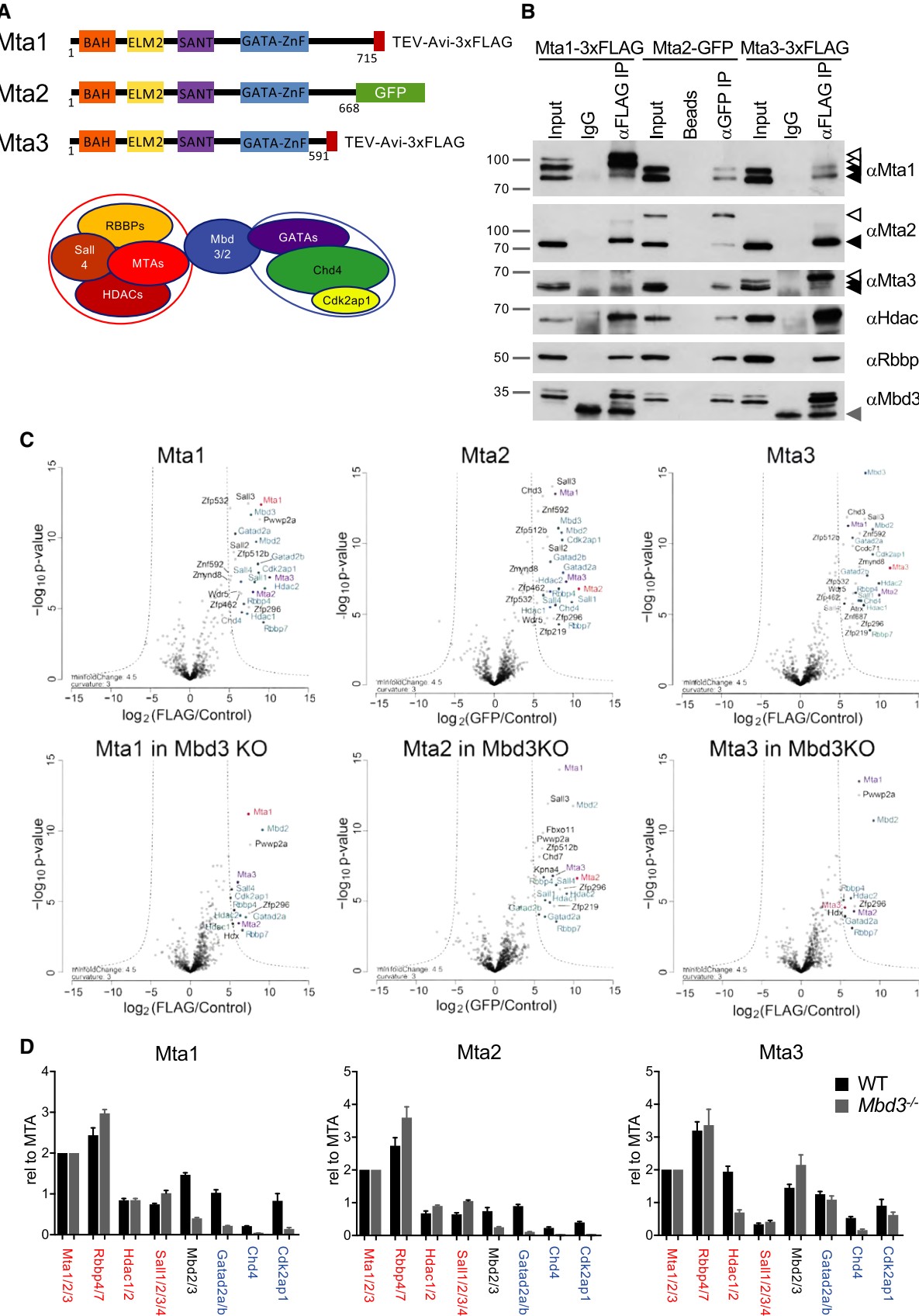

**Figure 1.**

◀

**Figure 1.   Biochemical characterisation of epitope-tagged MTA proteins.**

A   Schematic of MTA proteins with different protein domains indicated as coloured boxes, and the C-terminal epitope tags indicated. The model below shows the MTA- and HDAC-containing subunit joined to the CHD4-containing subunit through the MBD protein.

B   Tagged MTA proteins in heterozygously targeted cell lines were immunoprecipitated and subsequent western blots probed with antibodies indicated at right. Solid black triangles indicate the locations of untagged proteins, and open triangles show the position of epitope-tagged proteins in the anti-Mta1, Mta2 or Mta3 panels; and a grey triangle shows the location of IgG bands in the anti-Mbd3 panel. "Beads" in the Mta2-GFP IP represents a mock immunoprecipitation using beads only, analogous to the "IgG" controls in the Mta1-3xFLAG and Mta3-3xFLAG IPs. 10% of the input was loaded into "Input" lanes. Molecular weights in kDa are shown at left. (Full western blot images are available in Mendeley Data.)

C   Proteins co-purifying with Mta1, Mta2 or Mta3 in IP-mass spectrometry experiments in wild-type cells (top) or Mbd3-null cells (bottom). Proteins showing significant enrichment with the bait protein are located outside the dotted lines. For all panels, the protein being immunoprecipitated is indicated in red, the other MTA proteins in purple, and other NuRD components in blue. Each IP/Mass spec experiment set comprised three independent immunoprecipitations from one nuclear extract preparation.

D   Relative enrichment of indicated proteins in MTA pulldowns from wild-type (WT) or *Mbd3*-null (Mbd3KO) ES cells, normalised to 2x MTA proteins. NuRD components comprising the remodelling subunit are labelled in blue, those comprising the deacetylase subunit in red. As MTA proteins are the bait in these experiments, they may be isolated more efficiently than their interaction partners. Error bars represent standard deviation from three (Mta1 and Mta2) or six (Mta3) replicate pulldowns.

cell lines to identify proteins interacting with each of the MTA proteins by label-free quantitative mass spectrometry. Each protein robustly co-purified with all known NuRD component proteins, including each of the MTA proteins, confirming that MTA proteins are not mutually exclusive within NuRD in ES cells (Fig 1C). In addition to NuRD components, each of the MTA proteins co-purified with Wdr5 as well as a number of zinc finger proteins, most of which had previously been identified as NuRD-interacting proteins (Bode *et al*, 2016; Spruijt *et al*, 2016; Ee *et al*, 2017; Matsuura *et al*, 2017).

As NuRD is assembled from a deacetylase subcomplex and a remodeller subcomplex joined through the Mbd3 protein (Fig 1A), loss of Mbd3 is expected to result in dissociation of these two subcomplexes. Endogenous tagging for each MTA protein was performed in an ES cell line harbouring a floxed *Mbd3* allele, and IP/Mass spectrometry was repeated in ES cells after *Mbd3* deletion in order to enrich for interactions specific for the deacetylase subcomplex. The majority of interactions with non-NuRD components was lost after *Mbd3* deletion, indicating that most of these proteins do not directly associate either with the MTA proteins or with the deacetylase subcomplex (Fig 1C). Exceptions to this were Zfp296, which was identified as interacting with all three MTA proteins in an *Mbd3*-independent manner, and Pwwp2a, which co-purified with Mta1 in both wild-type and *Mbd3*-null cells, consistent with recent reports (Link *et al*, 2018; Zhang *et al*, 2018). Notably, Zfp219 and Zfp512b both showed *Mbd3*-independent interactions specifically with Mta2 (Fig 1C).

By quantitating the abundance of peptides sequenced in each experiment, we found that the interactomes for both Mta1 and Mta2 showed a depletion of peptides associated with the remodeller subcomplex (i.e. Chd4, Gatad2a/b and Cdk2ap1) in *Mbd3*-null cells, whereas proteins associated with the histone deacetylase subcomplex (MTA proteins, Hdac1/2, Sall proteins and Rbbp4/7) remained present at similar levels (Fig 1D). In contrast, the Mta3 interactome showed an increased interaction with Mbd2 and no relative loss of either subcomplex in the absence of Mbd3. Thus, both Mta1 and Mta2 can form part of a stable histone deacetylase-containing subcomplex in the absence of intact NuRD, but Mta3 is preferentially found in intact NuRD complexes. All three MTA proteins associated with Mbd2 in the absence of Mbd3, indicating that they all can contribute to Mbd2/NuRD. Together these data show that the MTA proteins are found exclusively within the NuRD complex in ES cells.

**MTA proteins show subtle differences in genome-wide binding patterns**

To test whether MTA proteins confer differential chromatin binding to NuRD complexes, we subjected our ES cell lines expressing epitope-tagged MTA proteins to chromatin immunoprecipitation followed by high throughput sequencing (ChIP-Seq). The binding profiles of the three proteins were largely, but not completely over-lapping (Fig 2A and B). Mta3 binding was almost entirely associated with Mta1 and/or Mta2 binding, while 31% of Mta1 peaks and 22% of Mta2 peaks were not associated with any other MTA protein (Fig 2A and B). The overall efficiency of Mta3-3xFLAG ChIP-seq was lower than those for Mta1-3xFLAG or Mta2-GFP, possibly due to the lower levels of Mta3 present in ES cells, although replicates were highly correlated (Appendix Fig S1). While Mta3-bound sites can therefore be identified with high confidence, it remains formally possible that "Mta1-only" or "Mta2-only" sites also contain Mta3, albeit at levels beneath the detection threshold of our ChIP-seq. The vast majority of MTA peaks were also bound by Chd4, indicating that they represent NuRD-bound regions (Figs 2C and D, and EV2A and B).

Sites found associated with all three MTA proteins were highly enriched for Chd4 and Mbd3 binding, consistent with these being core NuRD binding regions (Fig 2D). These sites were also enriched for marks of active promoters (H3K4Me3, H3K27Ac) and active enhancers (H3K4Me1, H3K27Ac, P300; Fig 2D and E), both of which are hallmarks of NuRD-associated regions (Miller *et al*, 2016; Bornelöv *et al*, 2018). The same was true for sites bound by any two of the three MTA proteins (Fig EV2B). The majority of sites occupied by one MTA protein, but not the other two, were also occupied by Chd4 but to a lesser extent than is seen at core NuRD sites (Fig 2D and E). Sites associated with all three MTA proteins, presumably representing core NuRD-bound sites, were located both at transcription start sites and at distal locations, while sites bound by only one MTA protein were predominantly located distal to transcription start sites (Fig EV2C). Whereas Mta2-only and Mta3-only sites showed some enrichment for both H3K4Me1 and H3K4Me3, Mta1-only sites showed no enrichment for H3K4Me3 (Fig 2D and E). In all cases, MTA-only sites lacking Chd4, which could represent sites of binding by the histone deacetylase subcomplex only, showed characteristics of inactive enhancers, in that they were moderately enriched for H3K4Me1 and P300, but not for H3K4Me3, H3K27Ac or H3K36Me3 (Fig EV2D). None of the MTA-bound

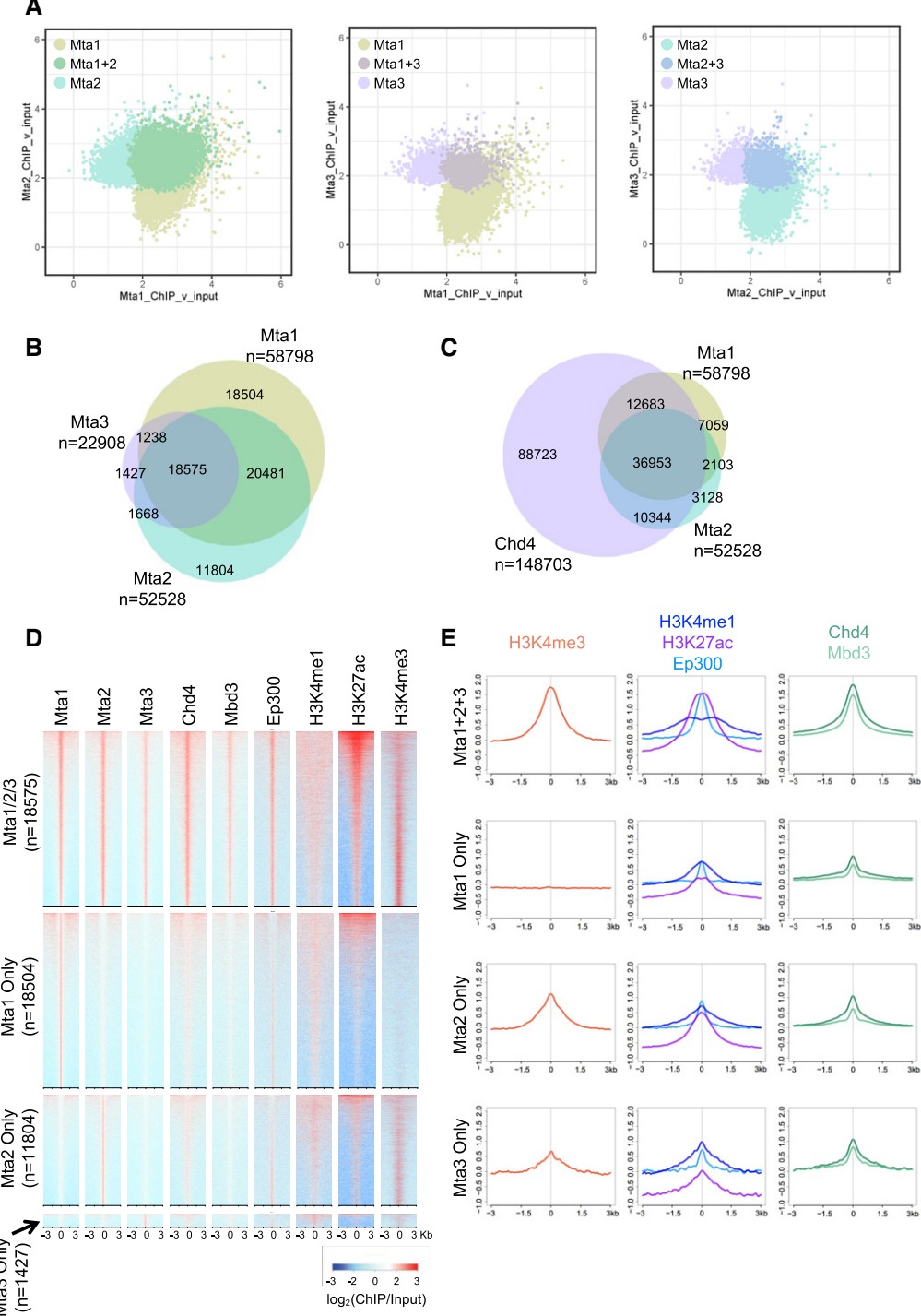

**Figure 2. MTA proteins show similar chromatin binding patterns.**

A ChIP-seq peaks identified for Mta1 and Mta2 (left), Mta1 and Mta3 (middle) or Mta2 and Mta3 (right) are plotted by enrichment for each protein. Peaks called for both proteins are indicated in the overlap.

B Overlap of peaks identified by ChIP-seq for each MTA protein in wild-type ES cells. Total peak numbers are indicated below each protein name. Each ChIP-seq dataset was made from biological triplicates.

C Comparison of Mta1 and Mta2 peaks with Chd4 peaks, as in Panel (B).

D ChIP-seq enrichment for indicated proteins or histone modifications is plotted across different subsets of Mta-bound sites. Mta1/2/3 refers to peaks identified in all three ChIP-seq datasets, whereas "Mta1 Only", "Mta2 Only" or "Mta3 Only" refer to peaks only called for that protein.

E Average enrichment of density plots in (D) is plotted for different subsets of Mta ChIP-seq peaks.

sequences showed any enrichment for methylated DNA (Fig EV2E). Genes associated with binding by only one or two MTA proteins, with or without Chd4, are associated with a similar distribution of GO terms (Fig EV2F) which is not consistent with the idea that the MTA proteins are directing NuRD activity to specific gene subsets.

## Mta1/Mta2/Mta3 triple knockout is a total NuRD null

To investigate the degree of genetic redundancy amongst the different *MTA* genes, we obtained gene trap alleles for each gene from the European Conditional Mouse Mutagenesis Programme (Skarnes *et al*, 2011) as ES cell lines (*Mta1* and *Mta2*) or as embryos (*Mta3*) (Fig EV3A–C). ES cell lines were used for morula aggregation to create chimaeric mice which were subsequently outcrossed to establish a mouse line. Conditional deletion alleles were generated for each line, which was subsequently bred with females expressing Sox2-Cre (Hayashi *et al*, 2002) resulting in deletion of floxed exons and the absence of any detectable protein production from each allele (Fig EV3D; see Materials and Methods for details). As has been reported previously, $Mta1^{-/-}$ and $Mta3^{-/-}$ mice were viable, and Mta2$^{-/-}$ mice showed incompletely penetrant embryonic lethality (Manavathi *et al*, 2007; Lu *et al*, 2008).

*Mta1*, *Mta2* or *Mta3*-null ES cell lines derived from mice were morphologically indistinguishable from wild-type ES cells, as were ES cell lines deficient for combinations of pairs of *MTA* genes (Fig EV3E). While some differences in gene expression could be identified between single mutants and their respective parent lines (Fig EV3F and G), these changes were not sufficient to make them distinct from wild-type ES cells when compared to *Mbd3*-mutant ES cells, 3.5 dpc ICM and 4.5 dpc epiblast (Fig EV3H). We therefore concluded that any gene expression changes identified in single mutant ES cells were unlikely to be of consequence during early development, and that MTA proteins show functional redundancy during early mammalian development.

$Mta1^{-/-}Mta2^{-/-}Mta3^{Flox/Flox}$ ES cells (*Mta12Δ*) were subsequently created and expanded in culture. After transfection with a Cre expression construct to induce deletion of both *Mta3* alleles, we recovered ES cells lacking all three MTA proteins (see Materials and Methods for details; Fig 3A). These $Mta1^{-/-}Mta2^{-/-}Mta3^{-/-}$ ES cells (subsequently referred to as *Mta123Δ*) appeared morphologically normal in standard, 2i + LIF culture (Fig 3B), but showed a considerable degree of spontaneous differentiation in serum/LIF culture. MTA proteins are therefore dispensable for ES cell viability.

Structurally, MTA proteins bridge an interaction between the deacetylase subcomplex with Mbd3 and the remodelling subcomplex (Fig 1A) so we predicted that loss of all three MTA proteins would prevent NuRD formation. Consistent with this prediction, we could detect no interactions between Chd4 and components of the deacetylase subcomplex (Hdac2, Rbbp4) in *Mta123Δ* ES cell nuclear extract by immunoprecipitation of endogenous proteins (Fig 3C). Surprisingly, despite being transcribed at normal levels, both Gatad2b and Mbd3 proteins were present at reduced levels in *Mta12Δ* cells and barely detectable in *Mta123Δ* cells (Figs 3C and EV4A), indicating that the MTAs are important for the stability of both of these proteins. To investigate this further, we monitored loss of protein stability in $Mta1^{-/-}Mta2^{-/-}Mta3^{Flox/Flox}$ ES cells after deletion using a tamoxifen-inducible Cre (Fig 3D). Loss of Mbd2, Mbd3, Gatad2a and Gatad2b protein stability was all coincident with loss of Mta3 protein, although Mbd3 and Gatad2b were already at reduced levels in the starting cell line ($Mta12ΔMta3^{Flox/Flox}$) (Fig 3D). Gatad2b and Mta3 also show reduced protein stability in *Mbd3Δ* ES cells, indicating that a particular inter-dependency exists between these three proteins for stability (Fig EV4B). Furthermore, introduction of either Mta1, Mta2 or Mta3 into *Mta123Δ* ES cells at levels comparable to wild-type expression resulted in restoration of Mbd3 protein levels, demonstrating that contact with at least one of the MTA proteins is sufficient for Mbd3 protein stability (Fig 3E). In contrast to *Mbd3*-null ES cells which display a significant depletion, but not a complete loss of NuRD due to partial compensation by Mbd2 (Kaji *et al*, 2006), *Mta123Δ* ES cells were completely devoid of any detectable intact NuRD (Fig 3C and F). The *Mta123Δ* ES cells are thus a total NuRD-null ES cell line, which allowed us to examine, for the first time, the consequences of a complete loss of the NuRD complex in a viable mammalian cell system.

## NuRD suppresses transcriptional noise

Twice as many genes showed an increase rather than a decrease in expression levels in *Mta123Δ* ES cells compared to control ES cells by RNA-seq (Fig 4A). The ratio of increased to decreased gene expression in *Mta123Δ* cells was very similar for genes bound by all three MTA proteins, Mta1+2, Mta1 only or Mta2 only (Fig 4B). NuRD's global impact on transcription is therefore not grossly altered by the inclusion or absence of individual MTA subunits.

Comparing global gene expression profiles using principal component analysis (PCA) showed that *Mta123Δ* and *Mbd3Δ* ES

---

**Figure 3. Mta1/Mta2/Mta3 triple-null ES cells represent a complete NuRD KO.**

A   Western blots of wild-type (Control), double or triple-null ES cell nuclear extract were probed with antibodies indicated at right. LaminB1 acts as a loading control. Approximate sizes are indicated in kDa.

B   Phase-contrast images of wild-type or *Mta123Δ* ES cells in 2i+LIF conditions. Scale bars represent 100 μm.

C   Western blots of anti-Chd4 immunoprecipitation of nuclear extract from indicated cell lines (top) probed with antibodies indicated at right. For the Gatad2b and Mbd3 blots, arrowheads indicate target proteins, while an asterisk marks a non-specific IgG bands. 10% of the input was loaded into "Input" lanes. Approximate sizes are indicated in kDa.

D   Western blot of a time course of *Mta3* deletion in $Mta1ΔMta2ΔMta3^{Flox/Flox}$: Cre-ER ($Mta12Δ3^{F/F}$) or Control ($Mta12Δ3^{F/F}$ without Cre-ER) ES cells probed for indicated NuRD component proteins or RNA polymerase II as a loading control (αRNAPII). The time course is indicated at the top as Days + tamoxifen.

E   Western blot showing rescue of *Mta123Δ* ES cells by ectopic expression of Mta1, Mta2 or Mta3 from a transgene (TG). Total RNA polymerase II acts as a loading control (αRNAPII). For all western blots, molecular weight is indicated at left in kDa.

F   Model of NuRD complex structure. Upon loss of Mbd3, most of the NuRD complex falls apart into the Chd4-containing remodeller subcomplex and the MTA-containing deacetylase subcomplex, but some intact Mbd2-NuRD still remains. Upon loss of all three MTA proteins, no intact NuRD remains, both Mbd3 and Gatad2a/b become unstable and neither of the intact subcomplexes remain.

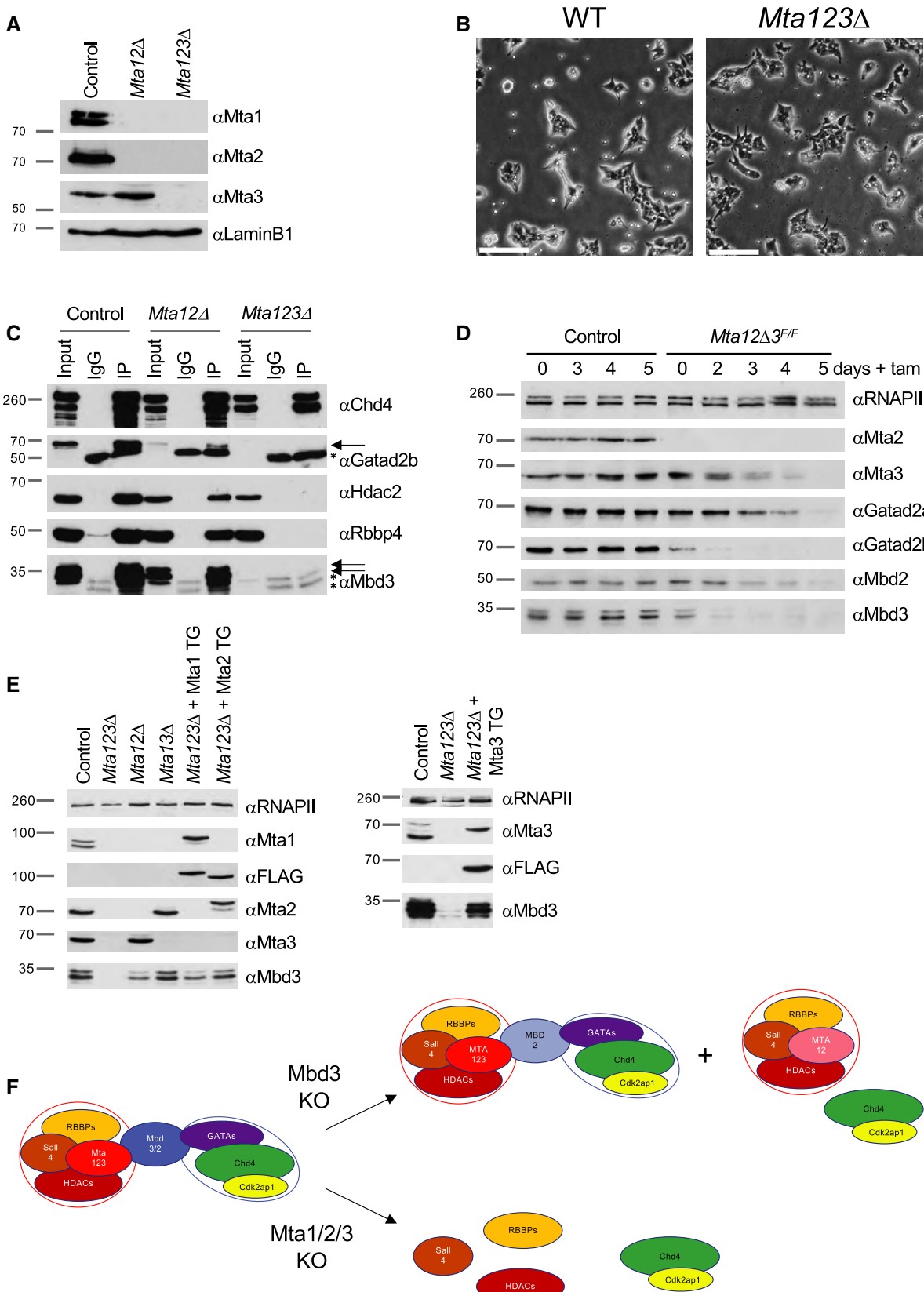

**Figure 3.**

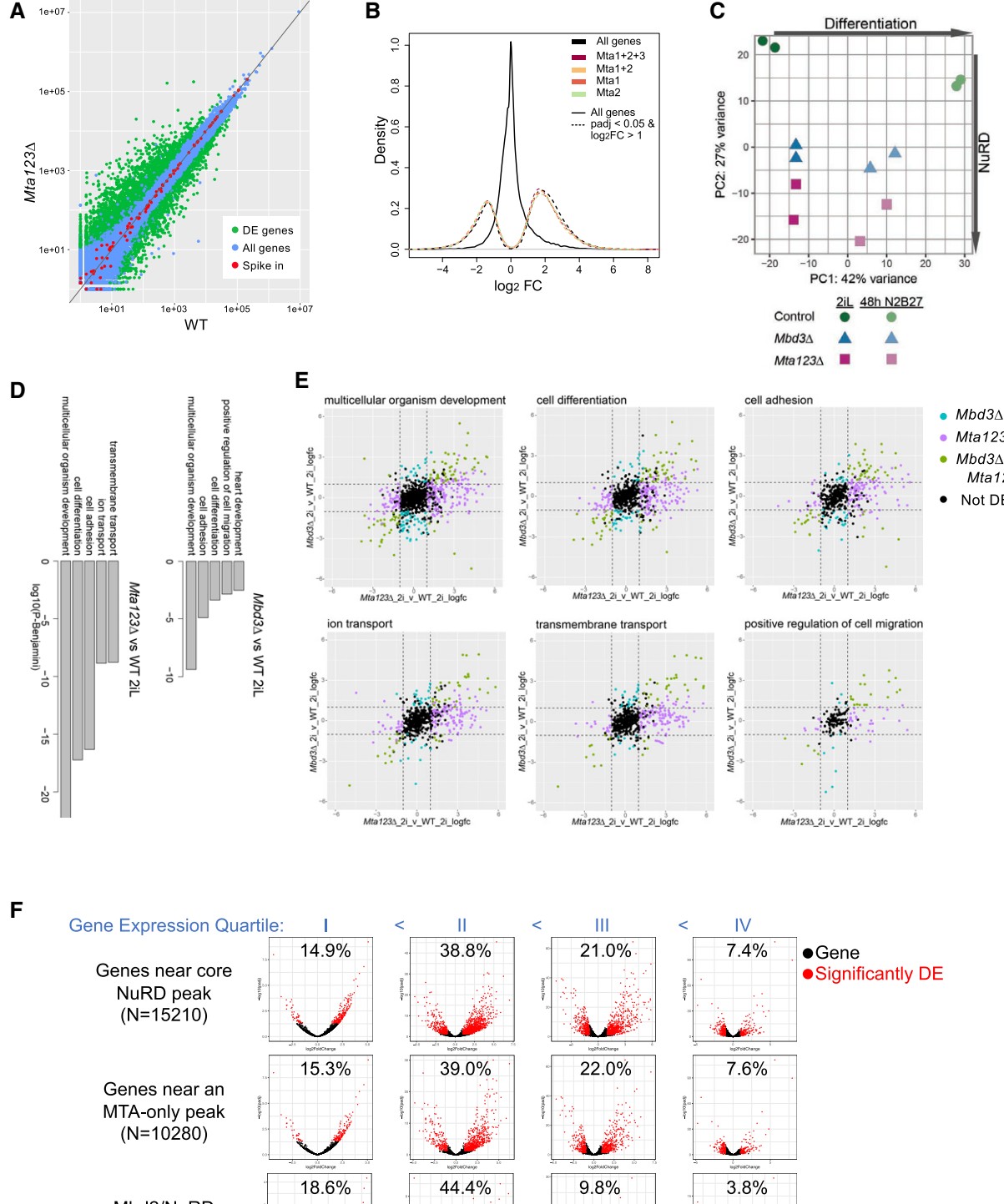

**Figure 4.**

**Figure 4.** MTA proteins act redundantly to control gene expression.

A Comparison of gene expression in *Mta123Δ* ES cells and wild-type (WT) ES cells. Red circles indicate spike-in controls, blue circles indicate genes that are not differentially expressed to a significant degree, and green circles indicate differentially expressed genes (2,404 increased, 1,293 decreased) defined with an adjusted *P*-value < 0.05 and a log2 fold-change > 1. *N* = 3 for each genotype.

B Fold-change in gene expression is plotted for different subsets of genes. All genes are plotted in black, while subsets of genes located nearest to ChIP-seq peaks for the indicated proteins +Chd4 and which show significant changes in expression compared to wild-type cells are plotted in dashed coloured lines as indicated. The number of genes in each of the Mta categories are as follows: all genes (32,271), all differentially expressed (*n* = 3,701), Mta1 + 2 + 3 (*n* = 1,738), Mta1 + 2 (*n* = 1,460), Mta1 (*n* = 1,020) and Mta2 (*n* = 924).

C Principal component analysis of RNA-seq data from ES cell lines of indicated genotypes in either self-renewing conditions (2iL) or after 48 h in the absence of two inhibitors and LIF (48 h N2B27). Each point represents a biological replicate.

D GO term enrichment for genes differentially expressed in *Mta123Δ* ES cells (left) or *Mbd3Δ* ES cells (right) compared to wild-type cells in 2iL conditions. For each comparison, the top five most significant gene ontology terms are plotted by $\log_{10}$ of the Benjamini-adjusted p-value. The significant GO terms and *P*-values were calculated using David v.6.8 (da Huang *et al*, 2009a).

E Genes associated with indicated GO terms plotted by fold-change in expression in *Mta123Δ* ES cells (*x*-axis) or *Mbd3Δ* ES cells (*y*-axis). Genes are coloured if they are differentially expressed (log2 fold-change > 1 and adjusted *P*-value < 0.05) in either comparison as indicated. The dotted lines show the fold-change cut-off of 2. GO terms were identified using David v.6.8 using a Benjamini score with a cut-off of 0.05.

F Genes associated with core NuRD peaks, with an MTA-only peak, with a Chd4 peak and an MTA-peak but not an Mbd3 peak (Chd4 + Mta-Mbd3 genes), or with an MTA-peak but not a Chd4 peak (MTA-only genes) were divided into quartiles based upon expression levels in wild-type cells. Change in gene expression is plotted on the *x*-axis and calculated significance of the expression change on the *y*-axis. Those genes significantly misexpressed ($|\log2FC| > 1$; $P \leq 0.05$) are indicated in red, and genes showing no significant change are plotted in black. The percentage of genes within each quartile which is significantly misexpressed is indicated at the top of each plot.

cells in self-renewal conditions were more similar to each other than either was to wild-type cells (Fig 4C). *Mta123Δ* ES cells were most distinct from wild-type cells, consistent with them representing a more complete NuRD knockout (Fig 4C). Consistent with this, genes driving the NuRD-specific separation (Principal Component 2; PC2) were generally misexpressed to a greater degree in *Mta123Δ* ES cells than in *Mbd3Δ* ES cells (Fig EV4C). Genes misexpressed in either *Mbd3Δ* or *Mta123Δ* ES cells were most significantly associated with the same top three GO terms, but the significance was greater in *Mta123Δ* ES cells as these cells misexpressed more genes in each category and generally to a greater extent than did the *Mbd3Δ* cells (Figs 4D and E, and EV4C).

Genes misexpressed in *Mta123Δ* ES cells were predominantly those normally expressed at lower levels (Fig EV4D). A very similar pattern was seen if we considered genes misexpressed in *Mbd3Δ* ES cells, with the major difference being that more genes were misexpressed in the *Mta123Δ* cells. The predominant biochemical difference between the *Mbd3Δ* cells and *Mta123Δ* cells was the lack of Mbd2/NuRD and of the MTA/HDAC subcomplex in the triple mutants, so we next asked whether we could detect evidence for specific functions for either of these complexes in ES cells. We identified genes near core NuRD peaks (*N* = 15,210), near Mta-only peaks (regardless of the presence of a NuRD peak; *N* = 10,280), genes associated with NuRD components Chd4 and MTAs, but not Mbd3 (presumptive Mbd2/NuRD-only genes; *N* = 349), and genes containing MTA-only peaks, but no Chd4 peak (MTA-only genes; *N* = 151) (Fig 4F) and divided them into expression quartiles. While there are very few genes not associated with a NuRD peak, those few presumptive Mbd2/NuRD-only genes or MTA-only genes showed a similar misexpression pattern to those seen for core NuRD genes, in that the majority of misexpressed genes was in the second lowest expression quartile (Fig 4F). We therefore find no evidence for a specific gene regulatory function associated with an MTA/HDAC subcomplex. This is consistent with a recent report in which loss of Pwwp2a/b resulted in only very slight alterations in nascent RNA production amongst a small subset of genes (Zhang *et al*, 2018). Nevertheless, this analysis further highlights the importance for NuRD in suppressing expression of lowly expressed genes. We

therefore conclude that, in addition to fine-tuning gene expression as part of the Mbd3/NuRD complex, MTA proteins are also important for preventing inappropriate activation of a variety of different genes, or in preventing transcriptional noise in ES cells.

**The NuRD complex safeguards cellular identity during differentiation**

We next asked what impact complete loss NuRD activity had upon the differentiation capacity of *Mta123Δ* ES cells. Upon removal of the two inhibitors and LIF from the culture media, wild-type cells began to adopt the flatter morphology of neuroectoderm (Fig 5A; Ying *et al*, 2003). This was accompanied by downregulation of pluripotency-associated genes and the activation of a neural gene expression programme (Fig 5B). *Mbd3Δ* ES cells are able to respond to the absence of self-renewal factors but have a very low probability of adopting a differentiated fate when induced to differentiate in N2B27 conditions (Kaji *et al*, 2006; Reynolds *et al*, 2012). Consistent with these findings, after 5 days in differentiation conditions *Mbd3Δ* ES cells showed some signs of having responded to differentiation conditions, but still retained pockets of morphologically undifferentiated cells (Fig 5A, Middle). In contrast, the completely NuRD-null *Mta123Δ* ES cells appeared to have all exited the self-renewal programme and adopted a flat, monolayer morphology (Fig 5A, Bottom). The ability of *Mta123Δ* ES cells to undergo morphologically normal neuroectodermal differentiation was rescued upon re-expression of either Mta1, Mta2 or Mta3 (Fig EV5A).

Since the differentiation process may be considered as a combination of exit from self-renewal (downregulation of pluripotency-associated genes) and acquisition of lineage-specific gene expression programmes, we focussed on changes to these two classes of genes by RT–qPCR over a differentiation time course. Pluripotency-associated genes such as *Esrrb*, *Nanog* and *Pou5f1* were downregulated during differentiation of wild-type, *Mbd3Δ* and *Mta123Δ* ES cells (Fig 5B). While this downregulation was not absolutely dependent on the presence of functional NuRD, the magnitude and kinetics of the response varied in a gene-dependent manner. Genes associated with acquisition of neural fate such as *Nestin*, *Pax6*, *Ascl1* and *Cdh2*

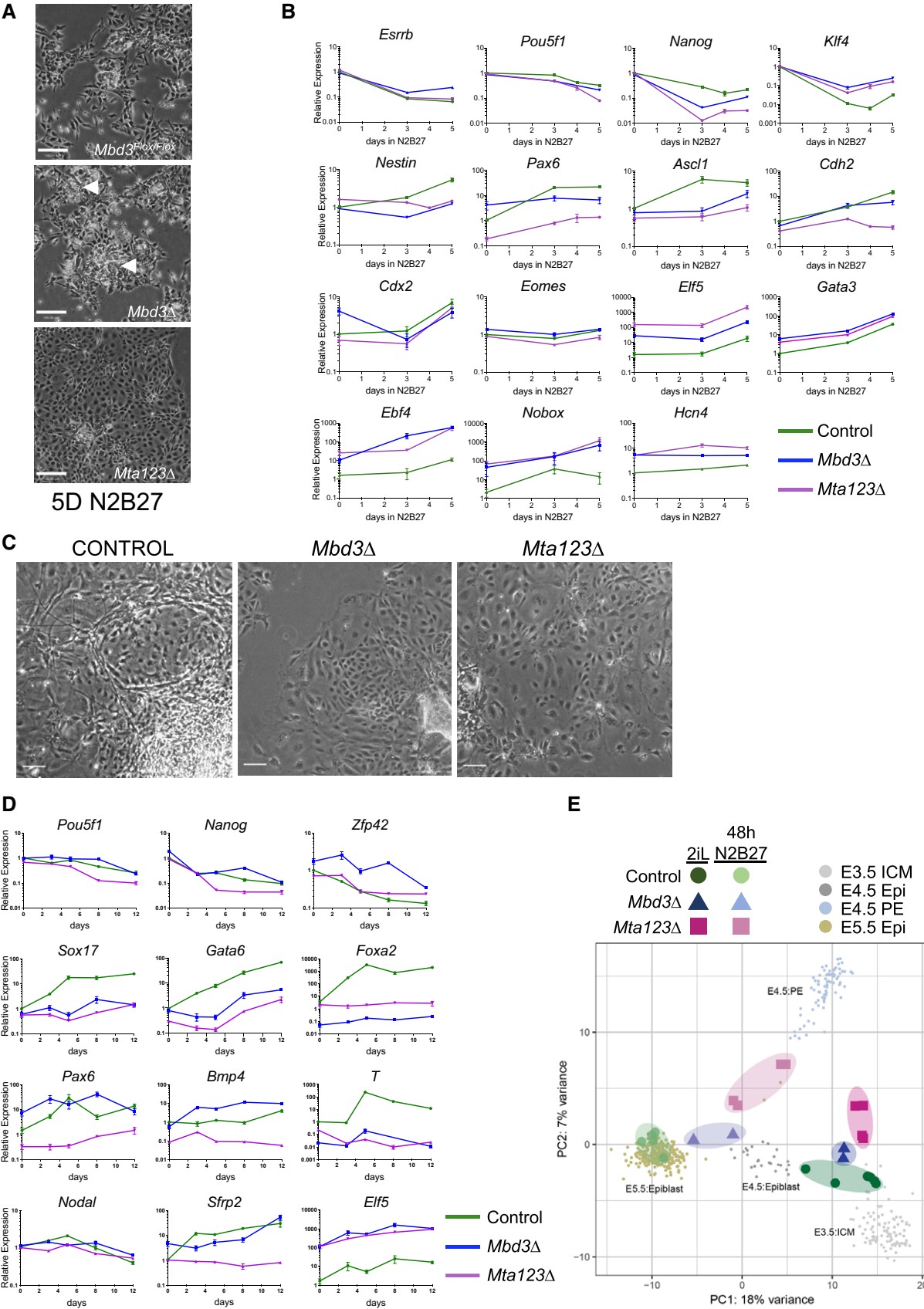

**Figure 5.**

**Figure 5.  NuRD activity maintains an appropriate ES cell differentiation trajectory.**

A    Phase-contrast pictures of ES cells of indicated genotypes after 5 days in neuroectoderm differentiation conditions (N2B27). White arrowheads in the *Mbd3Δ* panel indicate pockets of ES-like cells. Scale bars represent 100 μm.

B    Expression of indicated genes in the neuroectoderm differentiation time course was measured by RT–qPCR in Control (green) *Mta123Δ* (magenta) or *Mbd3Δ* (blue) ES cells. $N \geq 3$ biological replicates, error bars indicate SEM.

C    Phase-contrast images of embryoid body outgrowths of indicated genotypes after 12 days in differentiation conditions (N2B27 + 10% serum). Scale bars represent 100 μm.

D    Expression of indicated genes in embryoid body cultures over time was measured by RT–qPCR as in Panel (B).

E    Comparison of expression data for wild-type, *Mbd3Δ* or *Mta123Δ* ES cells in self-renewing (2iL) conditions or after 48 hours in differentiation conditions (48 h N2B27) with mouse embryonic single cell RNA-seq data from Mohammed *et al* (2017). Larger shading encloses biological replicates, and smaller circles represent individual cells. PC4 vs PC1 is plotted in Fig EV5F and a loading plot in EV5G.

were activated in the presence of wild-type NuRD, but this response was reduced or absent in the NuRD mutant lines (Fig 5B). In contrast, *Mta123Δ* ES cells aberrantly expressed markers of inappropriate lineages, such as *Elf5* and *Gata3* (trophectoderm—but not *Cdx2* or *Eomes*), *Ebf4*, *Nobox* and *Hcn4* (genes associated more generally with later stages of differentiation or transmembrane transport). Misexpression of these genes was also seen in *Mbd3Δ* cells, although not always to the same extent. Exogenous expression of either Mta1, Mta2 or Mta3 was able to rescue the ability of *Mta123Δ* ES cells to activate neural gene expression to different extents during differentiation (Fig EV5B), consistent with their ability to rescue morphological phenotypes.

The morphology of *Mta123Δ* cells induced to differentiate towards neuroectoderm was quite different from that of *Mbd3Δ* cells (Fig 5A), but there was no large-scale difference in gene categories changing between the two mutant lines after 48 h in differentiation conditions (Fig EV5C). In general, *Mta123Δ* cells misexpressed more genes from each category and to a greater extent than did *Mbd3Δ* cells (Fig EV5D), indicating that their abnormal morphology in differentiation conditions is most likely due to widespread gene misexpression generally, rather than activation of a specific differentiation pathway. *Mta123Δ* ES cells induced to differentiate towards a mesoderm fate similarly failed to appropriately activate differentiation markers, and exogenous expression of individual MTA proteins was again able to rescue the defect to varying degrees (Fig EV5E).

To assess the differentiation potential of *Mta123Δ* ES cells in an unbiased manner, we allowed control, *Mbd3Δ* and *Mta123Δ* ES cells to differentiate in embryoid bodies in basal media supplemented with serum (Doetschman *et al*, 1985). After 12 days, control cells had formed colonies containing numerous morphologically distinguishable cell types, silenced pluripotency-associated gene expression and activated expression of genes associated with all three germ layers (Fig 5C and D). In contrast, both *Mbd3Δ* and *Mta123Δ* cells showed less vigorous outgrowths consisting predominantly of a monolayer of flat cells (Fig 5C). While both mutant lines were able to silence expression of pluripotency-associated genes, both showed reduced activation of differentiation-associated genes, with *Mta123Δ* cells generally showing more profound expression defects than *Mbd3Δ* cells (Fig 5D). Together these data show that while NuRD is not strictly required for silencing of pluripotency genes, its activity is required for proper activation of lineage-appropriate genes and repression of lineage-inappropriate genes during differentiation.

To better understand how NuRD-null cells respond when induced to differentiate, we compared their gene expression profiles to a transcription landscape made from single cells taken from early

mouse embryos (Figs 5E and EV5F and G) (Mohammed *et al*, 2017). In the self-renewing state, *Mbd3Δ* and *Mta123Δ* cells clustered near control (parental) lines near embryonic day 3.5 (E3.5) and E4.5 inner cell mass cells, as is expected for naïve mouse ES cells (Boroviak *et al*, 2014). After 48 hours of differentiation, control ES lines clustered with E5.5 epiblast cells, indicating that the ES cell differentiation process occurred analogously to development *in vivo*. While *Mbd3Δ* ES cells appear to have exited the self-renewing state and to have taken the same differentiation trajectory as wild-type cells (i.e. leftwards along PC1; Figs 5E and EV5G), they did not travel as far along this trajectory as wild-type cells, instead occupying a space between the E4.5 and E5.5 epiblast states. *Mta123Δ* cells travelled even less far along PC1 than *Mbd3Δ* ES cells, and rather than maintain the appropriate differentiation trajectory they also travelled along PC2, occupying a space between E4.5 epiblast and E4.5 primitive endoderm. This further demonstrates that not only is NuRD important for cells to be able to adopt the appropriate gene expression programme for a given differentiation event, but it is also important for cells to maintain an appropriate differentiation trajectory.

If *Mta123Δ* ES cells were undergoing a specific trans-differentiation event towards trophectoderm during ES cell differentiation, then this could become more pronounced if exposed to normal differentiation conditions in a chimaeric embryo. If, in contrast, they are simply unable to differentiate properly, they would not be expected to contribute to early embryos. To distinguish between these possibilities, we assessed the ability of *Mta123Δ* ES cells to differentiate in chimaeric embryos. Equal numbers of control or *Mta123Δ* ES cells expressing a fluorescent marker were aggregated with wild-type morulae and allowed to develop for 48 hours. While wild-type cells contributed to the ICM of host embryos with 100% efficiency, *Mta123Δ* ES cells showed significantly reduced contribution and increased levels of apoptosis (Figs 6A and B, and EV5H). Those *Mta123Δ* cells that did survive in blastocysts were predominantly, but not always found in the inner cell mass. These cells expressed Sox2 but not Cdx2, indicating that they had not undergone inappropriate differentiation towards a trophectoderm fate (Fig 6A and B). Nevertheless, no *Mta123Δ* cells could be found in E5.5 dpc embryos, indicating that they were unable to adapt to developmental cues and form epiblast. This is most consistent with *Mta123Δ* cells being unable to enter a normal differentiation path in an ICM environment. The ability to contribute to the ICMs of chimaeric embryos was rescued by constitutive expression of Mta2 in *Mta123Δ* cells (Fig 6B). We therefore propose that NuRD not only functions to establish the correct lineage identity of cells during the differentiation process, but also

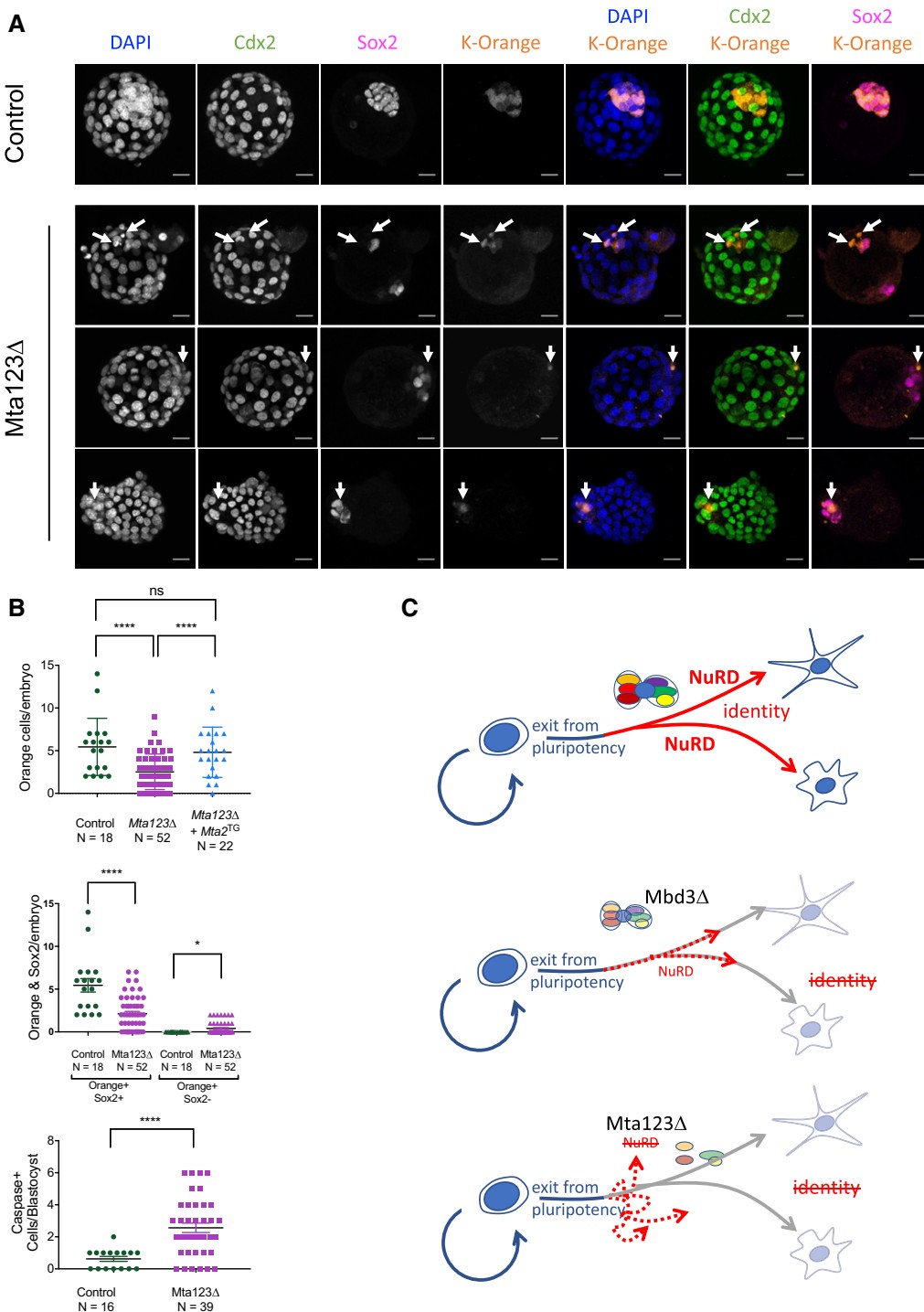

**Figure 6. NuRD-null ES cells cannot contribute to normal embryonic development.**

A  Composite images of representative chimaeric embryos made with control ($Mta1^{Flox/-}Mta2^{+/+}Mta3^{Flox/Flox}$) or $Mta123\Delta$ ES cells. ES-derived cells express the Kusabira Orange fluorescent marker. Sox2 indicates epiblast cells and Cdx2 is expressed in trophectoderm cells. Arrows indicate examples of K-Orange expressing cells in the mutant embryos. Scale bars = 20 μm.

B  (Top) Number of K-Orange expressing cells observed in chimaeric embryos obtained using control ES cells, $Mta123\Delta$ ES cells or $Mta123\Delta$ ES cells in which Mta2 was reintroduced on a constitutively expressed transgene ($Mta123\Delta + Mta2^{TG}$). P-values calculated using a two-tailed t-test. (Middle) Mean number of K-Orange cells per embryo separated by Sox2 expression. P-values calculated using a two-tailed t-test. (Bottom) Number of K-Orange and Caspase-3-positive cells per embryo. P-values calculated using a one-tailed t-test: *$P < 0.05$, ****$P < 0.0001$, "ns" = not significant.

C  Model of NuRD function during differentiation of pluripotent cells. NuRD facilitates lineage commitment of ES cells after exit from pluripotency (red arrows), allowing cells to form differentiated cell types (Top). In the absence of Mbd3, residual NuRD activity ensures cells retain the appropriate differentiation trajectory, but the cells are unable to reach a differentiated cell fate (dotted red arrows; Middle). In the absence of all three MTA proteins, there is no residual NuRD activity and ES cells are unable to either achieve appropriate lineage commitment, or to maintain the proper differentiation trajectory (dotted arrows, Bottom).

prevents inappropriate gene expression to maintain an appropriate differentiation trajectory (Fig 6C).

## Discussion

Here, we provide a biochemical and genetic dissection of the core NuRD component MTA proteins in mouse ES cells. In contrast to what has been found in somatic cell types, MTA proteins are not mutually exclusive in ES cell NuRD complexes and all combinations of MTA homo- and heterodimers can exist within NuRD. Different MTA proteins exhibit subtle differences in chromatin localisation or biochemical interaction partners, but we find no evidence for protein-specific functions in self-renewing or differentiating mouse ES cells. ES cells completely devoid of MTA proteins are complete NuRD nulls and are viable but show inappropriate expression of differentiation-associated genes, are unable to maintain an appropriate differentiation trajectory and do not contribute to embryogenesis in chimaeric embryos.

Protein subunit diversity is often found in chromatin remodelling complexes which specifies functional diversity (Hargreaves & Crabtree, 2011; Morey et al, 2012; Hota & Bruneau, 2016). This is also the case for the NuRD complex, where alternate usage of Mbd2/3, Chd3/4/5 and Mta1/2/3 has been found to result in alternate functions for NuRD complexes (Feng & Zhang, 2001; Fujita et al, 2003; Nitarska et al, 2016). Yet the findings by us and others (Manavathi et al, 2007 and Mouse Genome Informatics) that Mta1 and Mta3-null mice are viable and fertile, and that we detect no major differences in the abilities of different MTA proteins to rescue the Mta123Δ ES cell phenotypes indicate that MTA proteins exhibit considerable functional redundancy during development. The MTA protein family expanded from a single copy to three orthologues near the base of the vertebrate clade. Triplication of this protein family may have allowed early vertebrates to diversify NuRD function in specific somatic tissues, conferring a selective advantage. Our finding that MTA proteins are functionally redundant in early embryonic development indicates that the ancestral, essential function of the MTA proteins in early development has been retained by all three mammalian orthologues.

Different MTA proteins are capable of interacting with each other in ES cells, so how the mutual exclusivity reported in other cell types might be achieved is not clear. One possibility is that the variable inclusion in NuRD of zinc finger proteins, such as the Sall proteins in ES cells, could influence the MTA makeup of NuRD complexes. This class of variable NuRD interactors, which include Sall1/2/3/4, Zfp423 (Ebfaz), Zfpm1/2 (Fog1/2) and Bcl11b, interact with RBBP and/or MTA proteins via a short N-terminal motif (Hong et al, 2005; Lauberth & Rauchman, 2006; Lejon et al, 2011). Of this class of proteins, Sall1 and Sall4 are the most highly expressed in ES cells, and Sall4 can associate with all three MTA proteins (Fig 1C; Miller et al, 2016). In contrast, the Zfpm1 (Fog1) protein was shown to preferentially associate with Mta1 and Mta2, but not Mta3, in a somatic cell line (Hong et al, 2005). Hence, it is possible that different proteins using this N-terminal motif to interact with NuRD in different cell types could act to skew the proportion of different MTA proteins included in the NuRD complex.

This genetic dissection has revealed that NuRD component proteins display a large degree of inter-dependence for protein stability, as has been reported for other multiprotein complexes (Roumeliotis et al, 2017). Mta1 and Mta2 both contain two distinct RBBP interaction domains, while Mta3 lacks the C-terminal most RBBP interaction domain (Millard et al, 2016). The Mta1 and Mta2 proteins show minimal loss of stability in the absence of Mbd3, but Mta3 requires an interaction with Mbd3 to be completely stable (Fig EV4B; Bornelöv et al, 2018). It is possible that the additional interaction with an Rbbp protein confers stability to Mta1 and Mta2 in the absence of Mbd3. This would be consistent with our interpretation that Mta3 preferentially exists within an intact NuRD complex (Figs 1 and 2). Rbbp4/7 confers histone H3 binding to the NuRD complex, so Mta3-containing NuRD may bind chromatin less tightly than Mta1- and/or Mta2-containing NuRD.

We have used pluripotent cells lacking Mbd3 extensively to show that NuRD plays important roles in control of gene expression during early stages of exit from pluripotency *in vivo* and in culture (Kaji et al, 2006, 2007; Latos et al, 2012; Reynolds et al, 2012). *Mbd3*-null ES cells contain Mbd2/NuRD and thus represent a NuRD hypomorph, rather than a NuRD-null. ES cells lacking all three MTA proteins show no detectable NuRD formation (Fig 3), and therefore, we believe, represent a true NuRD null. *Mta123Δ* ES cells are similar to *Mbd3Δ* ES cells in that both misexpress a range of genes in 2iL conditions and both fail to properly differentiate. Yet the *Mta123Δ* ES cells misexpress more genes and generally to a greater extent than do *Mbd3Δ* cells in both self-renewing and differentiation conditions, and they show more pronounced differentiation defects. We propose that this phenotype is a direct consequence of NuRD's function in suppressing transcriptional noise during cell fate decisions. Moderate amounts of noise in the *Mbd3Δ* state allow cells to maintain a differentiation trajectory, but interfere with their ability to achieve the appropriate differentiation state (Fig 6C). In the higher levels of noise, characteristic of a complete absence of NuRD cells are unable to appropriately follow differentiation cues and maintain a coherent differentiation trajectory (Fig 6C). We therefore propose that NuRD encodes a highly conserved, redundant function to prevent transcriptional noise, as well as a further, Mbd3-dependent function to fine tune transcriptional outputs during differentiation.

## Materials and Methods

### Mouse embryonic stem cells

Mouse embryonic stem (ES) cells were grown on gelatin-coated plates in 2i/LIF conditions as described (Ying et al, 2008). All lines were routinely PCR genotyped and tested for mycoplasma. All ES cell lines used in this study are listed in Table EV1.

The following "Knockout First" alleles were obtained from EUCOMM as heterozygous ES cell lines (Illustrated in Fig EV1A and B):

Mta1[tm1a(EUCOMM)Wtsi]: https://www.mousephenotype.org/data/alleles/MGI:2150037/tm1a(EUCOMM)Wtsi

Mta2[tm1a(EUCOMM)Wtsi]: https://www.mousephenotype.org/data/alleles/MGI:1346340/tm1a(EUCOMM)Wtsi

Embryonic stem cells were used to derive mouse lines by blastocyst injection using standard methods. ES cell derivation was performed by isolating ICMs and outgrowing in 2i/LIF media as described (Nichols *et al*, 2009).

The epitope-tagged ES cell lines Mta1-Avi-3×FLAG and Mta2-GFP were generated by traditional gene targeting, while the Mta3-Avi-3×FLAG line was generated using a CRISPR/Cas9 genome editing approach. *Mta3* contains three alternate stop codons in exons 14, 16 and 17, with exon 15 accessed by alternate usage of a splice donor site in exon 14. Visual inspection of our RNA-seq data on a genome browser showed very little transcription beyond exon 14 in ES cells, indicating that the short isoform was predominant (Appendix Fig S2). We therefore targeted the stop codon present in exon 14 and simultaneously destroyed the alternate splice donor site using a synonymous mutation (GAA TGT to GAA TGC) to prevent splicing around our inserted construct in the targeted allele. The following sequences were targeted by guide RNAs in this experiment: 5′-G GCAAGGAGCGGAACGCGGA-3′ and 5′-GGATGGCAAGGAGCGGAA CG-3′. All MTA epitope-tagged lines were made in an *Mbd3*$^{Flox/-}$ background (Kaji *et al*, 2006).

*Mta123Δ* ES cells were made by targeting *Mta2* in *Mta1*$^{Δ/Δ}$ *Mta3*$^{Flox/Flox}$ ES cells. Guide RNAs targeting exons 2 and 18 of *Mta2* were transfected along with a targeting construct designed to replace exons 2–18 with a puromycin expression cassette. The resulting ES cell lines had replaced exon 2–18 with the selection cassette on one allele and had a deletion of the same exons on the other allele. The following sequences were targeted by guide RNAs: exon 2 (5′-TA GACGTAATCTGTAGGAGG-3′ and 5′-AAATAGACGTAATCTGTAG G-3′) and exon 18 (5′-AAATGCGCCGAGCGGCCCGA-3′ and 5′-TCAC CTGGAAATGCGCCGAG-3′).

Embryonic stem cells were induced to differentiate towards a neuroectoderm fate by removal of two inhibitors and LIF and culturing in N2B27 media as described (Ying *et al*, 2008). Mesendoderm differentiation was performed as follows: ES cells were plated at $10^4$ cells/cm$^2$ in N2B27 on fibronectin-treated 6-well plates and cultured for 48 h. Medium was then replaced with 10 ng/ml activin A and 3 μM CHIR99021 in N2B27 and cultured further. Embryoid bodies were allowed to form by allowing ~500 cells to aggregate in hanging drops of N2B27 media supplemented with 10% foetal calf serum. After 3 days, embryoid bodies were allowed to attach and outgrow on gelatin-coated plates in N2B27 + serum. Experiments were carried out in quadruplicate.

## Mice

All animal experiments were approved by the Animal Welfare and Ethical Review Body of the University of Cambridge and carried out under appropriate UK Home Office licences. The Mta3 "Knockout First" allele was obtained from EUCOMM as a heterozygous mouse line (Illustrated in Fig EV1C):

Mta3$^{tm3a(KOMP)Wtsi}$: (https://www.mousephenotype.org/data/allele s/MGI:2151172/tm3a%2528KOMP%2529Wtsi?).

Heterozygous knockout first mouse lines were crossed to a Flp-deleter strain kindly provided by Andrew Smith (University of Edinburgh; Wallace *et al*, 2007) to generate conditional alleles. Mice harbouring conditional alleles were crossed to a Sox2-Cre transgenic

line (Hayashi *et al*, 2002) to create null alleles. Mouse lines created in this study are listed in Table EV1.

Chimaeric embryos were made by morula aggregation with 8–10 ES cells per embryo as described (Hogan *et al*, 1994) and cultured for 24–48 h prior to fixation and immunostaining. ES cells used stably expressed a PiggyBac Kusabira Orange transgene.

## Nuclear extracts and immunoprecipitation

For extraction of nuclear proteins, ES cells were lysed in ice cold Buffer A (10 mM HEPES, pH7.9, 1.5 mM MgCl$_2$, 10 mM KCl and 0.5 mM DTT) supplemented with protease inhibitors and containing 0.6% NP-40. Nuclear pellets were washed further with Buffer A to reduce cytoplasmic protein contamination before extraction by vigorous shaking at 4°C for 1 hour in ice cold Buffer C (10 mM HEPES, pH7.9, 12.5% glycerol, 0.75 mM MgCl$_2$, 10 mM KCl, 0.5 mM DTT) with protease inhibitors followed by centrifugation to remove insoluble fraction.

For immunoprecipitation of FLAG-tagged proteins, anti-FLAG antibody was incubated with Protein G Sepharose beads (Sigma) for 1 h at room temperature. Nuclear extract (200 μg) was diluted in IP Buffer (50 mM Tris–Hcl, pH 7.5, 150 mM NaCl, 1 mM EDTA, 1% Triton X-100, 0.1% SDS, 0.1% DOC) in the presence of protease inhibitors and benzonase and incubated with antibody-bead conjugates at 4°C overnight. Beads were washed three times in ice cold IP Buffer and a further two times in IP Buffer containing 300 mM NaCl.

Chd4 immunoprecipitations were carried out at 4°C overnight using antibody to endogenous Chd4 with Protein G sepharose in 50 mM Tris–Hcl, pH 7.5, 150 mM NaCl, 1 mM EDTA, 1% Triton X-100, 10% glycerol and washed as for the FLAG IPs before SDS–PAGE.

For immunoprecipitation of GFP-tagged proteins, Chromotek GFP-Trap agarose beads were used. Nuclear extract (200 μg) was incubated with 25 μl GFP-Trap beads in GFP IP buffer (10 mM Tris, pH 7.5, 150 mM NaCl, 0.5 mM EDTA) with Benzonase and protease inhibitors for 1 h at 4°C. Beads were washed five times in IP buffer prior to western blotting.

IPs were carried out in bulk and loaded on multiple SDS gels for western blot analysis with various antibodies. Antibodies are listed in Table EV2.

## Label-free pulldowns and label-free quantitation (LFQ) LC-MS/MS analysis

Label-free GFP pulldowns were performed in triplicate as previously described (Kloet *et al*, 2018). 2 mg of nuclear extract was incubated with 7.5 μl GFP-Trap beads (Chromotek) or 15 μl FLAG-sepharose beads (Sigma) and 50 μg/ml ethidium bromide in Buffer C (300 mM NaCl, 20 mM HEPES/KOH, pH 7.9, 20% v/v glycerol, 2 mM MgCl$_2$, 0.2 mM EDTA) with 0.1% NP-40, protease inhibitors and 0.5 mM DTT in a total volume of 400 μl. After incubation, 6 washes were performed: 2 with Buffer C and 0.5% NP-40, 2 with PBS and 0.5% NP-40, and 2 with PBS. Affinity purified proteins were subject to on-bead trypsin digestion as previously described (Smits *et al*, 2013). In short, beads were resuspended in 50 μl elution buffer (2 M urea, 50 mM Tris, pH 7.5, 10 mM DTT) and incubated for 20 min in a thermoshaker at room temperature. After addition of 50 mM iodoacetamide (IAA), beads were incubated for

10 min in a thermoshaker at room temperature in the dark. Proteins were then on-bead digested into tryptic peptides by addition of 0.25 μg trypsin (Promega) and subsequent incubation for 2 h in a thermoshaker at room temperature. The supernatant was transferred to new tubes and further digested overnight at room temperature with an additional 0.1 μg of trypsin. Tryptic peptides were acidified and desalted using StageTips (Smits *et al*, 2013) prior to mass spectrometry analyses.

### Chromatin immunoprecipitation (ChIP), sequencing and analysis

Chromatin immunoprecipitations were carried out as previously outlined (Reynolds *et al*, 2012). For sequencing of ChIP DNA, samples from six (Mta1-FLAG and Mta2-GFP) and four (Mta3-FLAG) individual ChIP experiments were used. Antibodies used are listed in Table EV2. ChIP-seq libraries were prepared using the NEXTflex Rapid DNA-seq kit (Illumina) and sequenced at the CRUK Cambridge Institute Genomics Core facility (Cambridge, UK) on the Illumina platform. Low-quality reads and adaptor sequences were removed by *Trim Galore!* (version: 0.4.1, https://github.com/FelixKrueger/TrimGalore). The trimmed ChIP-seq reads were aligned to the mouse reference genome (GRCm38/mm10) using *bowtie* (version: 0.12.8; options: -m 1 -v 1; Langmead, 2010). This was followed by peak calling using *macs2 callpeaks* against the corresponding inputs using default parameters and a Q-value of <0.05. The R/Bioconductor *Diffbind* package was used to compare each combination of bound MTA proteins against all others using cut-off values of RPKM > 1.5, logFC > 1 and FDR < 0.05. The peaks were processed and visualised by *deepTools computeMatrix* and *plotHeatmap* (Ramirez et al, 2014). Venn diagrams were plotted using the R/Bioconductor *ChIPseeker* package.

DNA Methylation data were obtained from Shirane *et al* (2016), low-quality reads and adaptor sequences were removed by *Trim Galore!* (version: 0.4.1), and reads were aligned to the mouse reference genome (GRCm38/mm10) using *Bismark* (version: 0.20.0; Krueger & Andrews, 2011). Alignments were de-duplicated and further processed using *Methpipe*. Only CpGs with a coverage of at least 3 reads were considered. CpG methylation levels were visualised by *deepTools* using a sliding window of 250 bp to give a smoothed track for the heatmaps and profiles. Sequencing datasets are listed in Table EV3.

### Gene expression analyses

Total RNA was purified using RNeasy Mini Kit (Qiagen) including on-column DNase treatment. First-strand cDNA was synthesised using SuperScript IV reverse transcriptase (Invitrogen) and random hexamers. Quantitative PCRs (qPCRs) were performed using TaqMan reagents (Applied Biosystems) on a QuantStudio Flex Real-Time PCR System (Applied Biosystems) or a StepOne Real-Time PCR System (Applied Biosystems). Gene expression was determined relative to housekeeping genes using the $\Delta C_t$ method. TaqMan assays or PCR primers are listed in Table 1.

### RNA-seq

Libraries for sequencing were prepared using the NEXTflex Rapid Directional RNA-seq kit (Illumina) or SMARTer® Stranded Total RNA-Seq Kit v2—Pico Input Mammalian (Takara Bio) and

**Table 1. List of TaqMan assays and primers used for gene expression analyses.**

| Gene | TaqMan assay |
|---|---|
| *Ascl1* | Mm03058063_m1 |
| *Atp5a1* | Mm00431960_m1 |
| *Bmp4* | Mm00432087_m1 |
| *Cdh2* | Mm01162497_m1 |
| *Cdx2* | Mm01212280_m1 |
| *Esrrb* | Mm00442411_m1 |
| *Eomes* | Mm01351985_m1 |
| *Elf5* | Mm00468732_m1 |
| *Foxa2* | Mm01976556_s1 |
| *Gapdh* | Mm99999915_g1 |
| *Gata3* | Mm00484683_m1 |
| *Gata6* | Mm00802636_m1 |
| *Klf4* | Mm00516104_m1 |
| *Nanog* | Mm02019550_s1 |
| *Nestin* | Mm00450205_m1 |
| *Nodal* | Mm00443040_m1 |
| *Pax6* | Mm00443081_m1 |
| *Pou5f1* | Mm03053917_g1 |
| *Ppia* | Mm02342430_g1 |
| *Sox1* | Mm00486299_s1 |
| *Sox17* | Mm00488363_m1 |
| *T* | Mm00436877_m1 |
| *Tbp* | Mm00446971_m1 |
| *Zfp42* | Mm03053975_g1 |

| Gene | Forward primer (5′–3′) | Reverse primer (5′–3′) |
|---|---|---|
| *Ebf4* | AACTGCGGGTCCTTATGTTC | TTTGTCTTTTCCGTCCCAGG |
| *Hcn4* | CCCGCCTCATTCGATACATT | AGCAGAAGCATCATGCCAAT |
| *Nobox* | CCTACGGAGAAGCTCTGCAA | TCTGGGGGAGAACAACCTTC |
| *Sfrp2* | AATGAGGACGACAACGACAT | ACGCCGTTCAGCTTGTAAAT |

sequenced on the Illumina platform as for ChIP-seq libraries. Low-quality reads and adaptor sequences were removed by *Trim Galore!* (version: 0.4.1), and the trimmed reads were aligned to mouse reference genome (GRCm38/mm10) using *Tophat* (version v2.1.0; Kim *et al*, 2013). Read counts per gene were obtained using *featureCounts* (version 1.5.0; Liao *et al*, 2014) with annotations from Ensembl release 86 (Yates *et al*, 2016). Normalisation, with and without spike-ins, and differential expression analyses were performed using the R/Bioconductor Deseq2 package (version 1.14.1; Love *et al*, 2014). The *R prcomp* function was used for principal component analysis (PCA). Gene Ontology (GO) analysis was performed using DAVID (da Huang *et al*, 2009b).

### Statistical analyses

No power calculations were performed to predetermine sample sizes. Statistical analyses were performed using GraphPad Prism

software. Images of cells or embryos are representative of at least three independent experiments. Statistical methods used to analyse high throughput sequencing data and proteomics data are described in the respective sections of the Materials and Methods.

## Data availability

High throughput sequence datasets used in this manuscript are listed in Table EV3. ChIP-seq data are available with the GEO accession number GSE122833 (https://www.ncbi.nlm.nih.gov/geo/query/acc.cgi?acc=GSE122833). RNA-seq data are available with the GEO accession number GSE122696 (https://www.ncbi.nlm.nih.gov/geo/query/acc.cgi?acc=GSE122696). The mass spectrometry proteomics data have been deposited to the ProteomeXchange Consortium via the PRIDE partner repository with the dataset identifier PXD009855 (https://www.ebi.ac.uk/pride/archive/projects/PXD009855). All original western blot images are available at Mendeley Data: https://doi.org/10.17632/d25bgfn8hd.1.

**Expanded View** for this article is available online.

## Acknowledgements
We thank Bill Mansfield, Peter Humphreys, Maike Paramor, Vicki Murray and Sally Lees for technical assistance and advice, and Alexander Brehm, Austin Smith, Ernest Laue and members of the BDH laboratory for discussions and comments on the manuscript. Funding to the BH and MV laboratories was provided through EU FP7 Integrated Project "4DCellFate" (277899). The BH laboratory further benefitted from a Wellcome Trust Senior Fellowship (098021/Z/11/Z) and from core funding to the Cambridge Stem Cell Institute from the Wellcome Trust and Medical Research Council (097922/Z/11/Z and 203151/Z/16/Z). The Vermeulen laboratory is part of the Oncode Institute, which is partly funded by the Dutch Cancer Society (KWF).

## Author contributions
TB and BH devised the study; TB, SK, SG, RF, JC, NR and BH generated the data; MB, MR and SD analysed high throughput sequencing data; SK and MV generated and analysed proteomics data; MK provided methodology; and NR and BH wrote the manuscript with input from other authors.

## Conflict of Interest
The authors declare that they have no conflict of interest.

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
