## [Review Process File · The EMBO Journal]

The Nucleosome Remodelling and Deacetylation complex suppresses transcriptional noise during lineage commitment

Thomas Burgold, Michael Barber, Susan Kloet, Julie Cramard, Sarah Gharbi, Robin Floyd, Masaki Kinoshita, Meryem Ralser, Michiel Vermeulen, Nicola Reynolds, Sabine Dietmann and Brian Hendrich

Review timeline:

Submission date:	25th Sep 2018
Editorial Decision:	12th Nov 2018
Authors' correspondence:	15th Nov 2018
Revision received:	15th Feb 2019
Editorial Decision:	22nd Mar 2019
Revision received:	26th Mar 2019
Editorial Decision:	29th Mar 2019
Revision received:	30th Mar 2019
Accepted:	2nd Apr 2019

Editor: Daniel Klimmeck

Transaction Report:

1st Editorial Decision

12th Nov 2018

Thank you for the submission of your manuscript (EMBOJ-2018-100788) to The EMBO Journal. Your manuscript has been sent to three referees, and we have received reports from all of them, which I enclose below.

As you will see, the referees acknowledge the potential high interest and novelty of your work, although they also express a number of issues that will have to be addressed before they can support publication of your manuscript in The EMBO Journal. Referee #1 states that the findings generality of the results remains unclear and thus asks you to investigate 21 culture conditions (ref#1, pt.2). Further, this reviewer requests expansion of the concept to somatic cells (ref#1, pt.1). In addition, referee #1 asks you to consolidate the *in vivo* differentiation assays and points to inconsistencies in the data (ref#3, pts.3,6). Referee #3 requests clarification of the individual contribution of MTAs to the downstream transcriptional programs, and in addition states issues related to claims not sufficiently supported by the biochemical data. Further, issues related to documentation of methodologies as well as missing controls would need to be conclusively addressed to achieve the level of robustness needed for The EMBO Journal.

I judge the comments of the referees to be generally reasonable and given their overall interest, we are in principle happy to invite you to revise your manuscript experimentally to address the referees' comments. I agree that achieving a more detailed picture of the individual MTA's contributions to downstream expression would significantly strengthen the study.

Please note that while *per se* well taken, referee#1's point on testing somatic cells is in our view beyond the scope of the current work.

I would appreciate if you contact me shortly regarding the referee's requests, in particular on the single KO analyses. Please let me know any time if you have additional questions or need further input on the referee comments.

REFeree REPORTS:

Referee #1:

Burgold et al. show in this manuscript that the NuRD components MTA1, 2 and 3 are not mutually exclusive in 2i-cultured ESCs cells using biochemical approaches, chromatin binding and knockout (KO) studies. They further show that Mta1/2/3 KO ESCs eliminate NuRD function more completely than Mbd3 KO ESCs. The main conclusion of this study is that NuRD suppresses transcriptional noise in undifferentiated ESCs and during differentiation. Consistent with this notion, Mta1/2/3 KO ESCs cells differentiated less efficiently than wild type cells. Overall, this is a well-performed and well-written study that makes a number of interesting observations on the role of the NuRD complex in the maintenance of, and exit from, pluripotency and is therefore in principle suitable for publication if the authors can address the following questions.

Major points:

1. A major conclusion of this study is that MTA proteins are not mutually exclusive within the NuRD complex in 2i-cultured ESCs, which differs from previous studies in somatic cells. This manuscript would be strengthened if the authors could test whether this observation extends (or not) to somatic cells in their own hands, for example by differentiating their triple KO line to fibroblast-like cells.
2. The authors conclude that MTA proteins and thus NuRD activity is dispensable for the maintenance of ESCs cultured in 2i. It would be informative to assess whether NuRD is also dispensable in ESCs cultured in conventional serum/LIF conditions, considering that certain genes are essential in serum/LIF-cultured ESCs but not in 2i-cultured ESCs (e.g., Prdm14).
3. It is surprising that Mta1/2/3 KO ESCs lead to the upregulation rather than downregulation of genes even though MTA proteins bind to active promoters/enhancers marks. How do the authors explain this apparent discrepancy? Are the transcriptional changes mostly due to secondary effects?
4. This manuscript would benefit from a bit more analysis and discussion of specific the genes/pathways that change in Mta1/2/3 KO ESCs compared to Mbd3 KO ESCs and controls. For example, what genes and functional gene categories drive the shift of Mta1/2/3 KO cells in the PCA plot in Fig. 5A and are these genes all direct targets of MTA proteins? Similarly, are the highlighted genes in Fig. 4G direct targets of MTA proteins?
5. The in vitro differentiation assays plus analysis (Fig. 4G) should be repeated with Mbd3 KO ESCs as a relevant control. Wouldn't the authors expect a more pronounced defect in exit from pluripotency in Mta1/2/3 KO ESCs based on previous work on Mbd3 KO ESCs by the authors and others?
6. The in vivo differentiation assay (ICM incorporation) of Mta1/2/3 KO ESC is difficult to interpret. I encourage the authors to perform teratoma analysis of Mta1/2/3 KO ESCs and compare results to Mbd3 KO ESCs and WT ESCs as an unbiased, spontaneous in vivo differentiation assay.

Minor point:

The color key in Fig. 5A is hard to see and should be changed

Referee #2:

Hendrich:

The current manuscript reports a highly comprehensive and detailed analysis of the function of Mta 1 to 3 in mouse stem cells. The detailed and thorough genetic and biochemical analysis adds important insights into the function of the NuRD complex. The set of Mta deletions represents for the first time a functional NuRD complex in mouse stem cells and the interpretation that this complex functions in reducing transcriptional noise is compelling.

Given 1. the overall relevance of the NuRD complex (as an important remodelling complex which is furthermore controversially debated in stem cell biology and reprogramming) and 2. the important general question if and how remodeler subunits mediate specific or redundant function this paper appears to be a strong candidate for EMBO journal after minor revision.

Points of criticisms:

Figure 2:

While I agree with the conclusion that Mta1-3 bind largely in a similar manner the authors present pie charts that are potentially misleading as they give the false impression of distinct targets.

Notably peaks that appear unique to one of the three proteins appear to have weak signal. A better comparison of the ChIPseq data is to provide pairwise scatterplots. These will likely reveal that "unique" peaks are mostly sites where the signal is barely below or above thresholds that is set by the peak recognition algorithm. Really unique sites should have strong signal in one protein and weak to no in another. Otherwise these can simply reflect different chip efficiencies.

Minor points:

Line 133 "peri"

Referee #3:

Burgold et al EMBO J

This manuscript explores the genetics and biochemistry of the MTA subunits of the NuRD complex. The authors perform biochemical and molecular experiments in ES cells to complement developmental experiments in embryos.

In general, the manuscript addresses a topic of interest to the chromatin community. The principal conclusion of the work is that MTA proteins act in a redundant fashion in ES cells to fine tune gene expression for development.

I find aspects of the biochemistry experiments to be incompletely described (see below) and other portions to be of poor quality.

The gene expression experiments focus on triple KO cell lines without directly testing the central conclusion of the manuscript - that the MTA proteins are redundant in terms of gene regulation. It seems to me a direct test would be to assess the transcriptional program in the single KO ES cells. From a high level perspective, it would be useful to the readers of EMBO J for the authors to discuss why evolution has resulted in three MTA paralogs in vertebrates of their function(s) are redundant.

Specific comments to assist the authors in improving their manuscript:

Line 138 Each MTA protein was able to immunoprecipitated both of the other MTA proteins in addition to unmodified forms of itself.

The data does not support this conclusion. I see no evidence in 1B that MTA1 can IP MTA3.

- Please provide a method for nuclear extract production for this experiment. Are the co-IPs sensitive to nuclease treatment?
- Data for MTA3 IP with MTA2-GFP is of poor quality. Please use a light chain specific secondary antibody (likewise coIP of HDAC2 and Rbbp4). The current data are not acceptable.
- It would help the reader to interpret this data if the authors could indicate what percentage of input

material is run for each IP in the Figure Legend.

IP Mass Spec

The methodology for the IP mass spec is also not described. Reference to a previous publication is not sufficient. Please describe the experiment

- Are the IP's sensitive to nuclease treatment? Ethidium bromide in the IP (as described at PRIDE) perturbs B form DNA helical structure but in no way changes the fact that DNA is a polymeric anion capable of bridging otherwise discrete particles.
- I do not understand the volcano plot and its description on the PRIDE site. Threshold adjustment so that control values are not significant. This is unclear. I generally believe p value is related to hypothesis testing. How do control values test a hypothesis and get a p value? Please explain how this works. Are the p values adjusted for multiple testing?
- I am unclear on the n for each experiment as reported in the evidence table on the PRIDE site. For example, there appear to be 6 separate experiments for MTA2-GFP mass spec, with 3 experiments for MTA2-GFP in MBD3 KO. There appear to be 3 experiments for MTA1 (all three conditions) and 6 for MTA3 (all conditions). I am unclear on why this is the case. Can one legitimately compare across analyses performed on different numbers of experiments? Doesn't this impact hypothesis testing? Does this signify that IP efficiency is vastly different with the different antibodies (FLAG v GFP)? Please clarify.
- What is the difference in the evidence table between MTA2_nonGFP and MTA2_WT in the evidence table on the PRIDE site?
- By my very simple analysis of the evidence table at PRIDE, it appears that MTA1 IP brings down much more MTA1 than MTA2 (572 peptides v 414) and substantially more MTA1 than MTA3 (572 peptides v 192). One can make similar observations for the other pulldowns - leading to the likelihood that MTA proteins may be more likely to IP themselves than their paralogs. Please provide analysis to address this - the data are present here and the analysis would be of interest to readers of EMBO J.
- The identification of CHD4 in the mass spec is somewhat surprising - identification of CHD4 is really low. GATAD2A/B are recovered with something like expected frequency. Do the authors believe that the nearly 10-fold difference in abundance of a putative positive control is indicative of a meaningful experiment? Please explain. Readers of EMBO J would benefit from discussion of this point. Likewise, the structural data from the Schwabe group would predict a 1 to 1 molar ratio of MTA protein with HDAC. It seems more like 2 to 1 for MTA1 and MTA2. Please explain for the reader (and for this reviewer - I do not understand).

ChIPseq

- I am unable to assess the quality of this data without access to genome browser compatible data. The authors need to make this data available for review.
- The ChIP data have very different peak numbers for MTA3 as compared to MTA1 and MTA2 (approximately 1/2 the number) - is this an outcome of lower expression or different chip quality (as I do not have access to the chip data, I am unable to discern quality...)
- It would be informative to readers to see the distribution relative to Refseq Transcription start sites of the MTA1/2/3 peaks and the MTA1, MTA2, MTA3 only peaks.
- The authors should report the QC and filtering information for the ChIPseq. Number raw reads, passing filter, deduplication, uniquely mapped, ...
- The reader would benefit from seeing the ChIP for the MTA proteins plotted as a scatter plot - MTA1 v MTA2, MTA1 v MTA3, MTA2 v MTA3. This type of plot permits the reader to not only see overlap, but also see magnitude of signal at overlapping regions.

MTA1,2,3 KO ES Cells

- Is MBD2 subject to the same type of protein-level control in the triple MTA KO ES cell line?
- It seems to me that GATAD2b abundance is greatly impacted in the MTA1,2 double KO. The bands in this blot are very confusing - heavy chain (?) is complicating the picture. Please guide the reader in the Figure legend to understand which band is the GATAD protein and which is antibody chain (?) in the IP. If, in fact, GATAD2B is lost in the MTA1,2 double KO, this merits discussion in the text. Is GATAD2A behaving similarly? What is the status of GATAD2A/B in the Mbd3 KO?
- It would be useful to see a gel filtration or sedimentation analysis of CHD4 in the MTA1,2,3 triple KO cell line to confirm and complement the coIP.

Gene Expression analysis

- Lines 251-255: this is a conclusion not supported by data. That a ratio of many genes is similar is not the same as no genes are impacted by the inclusion or absence of individual MTA subunits. This conclusion requires a different experiment. The current statement is true only at a global level in the absence of further experiments.
- The authors conclude that NuRD suppresses expression of lowly expressed genes. This analysis is based on fold change thresholds to define altered expression. This type of analysis is likely to bias towards low basal expression. Does analysis using a different metric (i.e. p value) lead to a similar conclusion?

Differentiation trajectory

- It is unclear to me what the basis of the conclusion drawn in lines 356-357. It looks to me in Figure 5B as though the MTA KO cells follow the same developmental path as control cells, there are just fewer of them. Please elaborate for the reader on how the data shown support the conclusion drawn.

Authors' correspondence

15th Nov 2018

Thanks for your email and for these reviews.

We do have the single KO ES cells. We had some data from these in a previous draft but removed them as they didn't really add anything. Nevertheless we can put them back in to address this reviewer's point.

Other comments should be relatively easily dealt with.

1st Revision - authors' response

15th Feb 2019

Please see next page.

Referee comments in BLACK

Our response in BLUE

Referee #1:

Major points:

1. A major conclusion of this study is that MTA proteins are not mutually exclusive within the NuRD complex in 2i-cultured ESCs, which differs from previous studies in somatic cells. This manuscript would be strengthened if the authors could test whether this observation extends (or not) to somatic cells in their own hands, for example by differentiating their triple KO line to fibroblast-like cells.

Evidence from mouse B-cells and human breast cancer cell lines indicates that in these tissues there is some degree of mutual exclusivity amongst Mta proteins in NuRD. In contrast, proteomics in HeLa cells shows that MTA proteins do interact in that cell line. In the text we point out these previous findings, making sure we do not give the impression that our results are at odds with these reports, but rather than in ES cells there is no mutual exclusivity. Defining the tissue distribution of MTA protein exclusivity would be certainly very interesting but is outside the scope of this study.

2. The authors conclude that MTA proteins and thus NuRD activity is dispensable for the maintenance of ESCs cultured in 2i. It would be informative to assess whether NuRD is also dispensable in ESCs cultured in conventional serum/LIF conditions, considering that certain genes are essential in serum/LIF-cultured ESCs but not in 2i-cultured ESCs (e.g., Prdm14).

Cells were transferred into serum/LIF conditions and maintained for several weeks with multiple passages. While control cells quickly adopted a homogeneous morphology of undifferentiated cells (left hand panel), *Mta123Δ* cultures consisted of a heterogeneous mixture of cells which persisted over multiple passages (right hand panel). In the image below one can see some cells which adopt the domed, closely packed morphology of ES cells, others which are flattened, reminiscent of epiSC, and others which are clearly differentiated. A similar array of cell types arise if the triple null is made in SL conditions (by inducing deletion of *Mta3* in *Mta1/2* double-null cells). Thus while triple null cells are viable in serum/LIF conditions, they are prone to differentiate and the cultures are very heterogeneous. We have mentioned this in the Results, but not included the figures in the manuscript.

Reviewer Figure. Control (left) and *Mta123Δ* ES cells (right) maintained for several passages in serum/LIF conditions. Scale bar represents 100μm.

3. It is surprising that *Mta1/2/3* KO ESCs lead to the upregulation rather than downregulation of genes even though MTA proteins bind to active promoters/enhancers marks. How do the authors explain this apparent discrepancy? Are the transcriptional changes mostly due to secondary effects?

This issue of how NuRD can both activate and repress transcription, and why it is present at all active enhancers and promoters was the subject of our recent paper (Bornelov et al., 2018). The expression volcano plots (Figs 4F and EV4D) show that the majority of genes showing misexpression in *Mta123Δ* cells are expressed at lower levels, but are still expressed. Thus while NuRD sits at all active promoters and enhancers, its activity has less of an impact on transcriptional output at highly expressed genes than it does at lowly expressed genes. At the latter it appears from this study that the predominant role of NuRD activity is to reduce transcription levels.

4. This manuscript would benefit from a bit more analysis and discussion of specific the genes/pathways that change in *Mta1/2/3* KO ESCs compared to *Mbd3* KO ESCs and controls. For example, what genes and functional gene categories drive the shift of *Mta1/2/3* KO cells in the PCA plot in Fig. 5A and are these genes all direct targets of MTA proteins? Similarly, are the highlighted genes in Fig. 4G direct targets of MTA proteins?

There are no dramatically different pathways changing in the *Mta123Δ* cells as compared to the *Mbd3Δ* cells: there are just generally more genes misexpressed in the triple mutant cells. To try to make this point more clear we now include the results of GO analyses on gene expression changes seen in *Mbd3Δ* and *Mta123Δ* cells during differentiation (Fig EV5C), as well as gene expression plots for the top GO terms (Fig EV5D). We also added in loading plots for Figure 5E in Figure EV5G, with a few key genes highlighted based upon a published classification of early embryonic gene expression (Boroviak et al., 2014). Although it appears the triple mutant cells are heading towards primitive endoderm in this plot, one must remember that this is a 2D projection of a very high dimensional dataset. This is why we also plot the data along PC1 and PC4 in Figure EV5F, which shows that the triple mutant cells do not resemble primitive endoderm along PC4. Rather they are not heading down any specific lineage path, as indicated by the GO plots in Fig EV5D. We have attempted to make this point more clear in the text.

As NuRD binds to virtually all promoters and enhancers, essentially all genes are direct targets of MTA proteins. So while we could provide UCSC screenshots with MTA ChIP-seq peaks at all genes for which we show qRT-PCR data, this would be somewhat disingenuous since we could also provide screenshots of lovely ChIP-seq peaks at genes that are not misexpressed in the mutants. How NuRD controls expression of some “targets” but not others is the focus of our recent paper (Bornelov et al., 2018).

5. The *in vitro* differentiation assays plus analysis (Fig. 4G) should be repeated with *Mbd3* KO ESCs as a relevant control. Wouldn't the authors expect a more pronounced defect in

exit from pluripotency in *Mta1/2/3* KO ESCs based on previous work on *Mbd3* KO ESCs by the authors and others?

We have now included *Mbd3*-null cells in the analysis of gene expression upon neural differentiation (Figure 5B) as well as during embryoid body differentiation (Figure. 5D). In general the *Mta123Δ* cells show a similar trend as do the *Mbd3Δ* cells, but often show a more pronounced defect in gene expression (e.g. *Pax6*, *Cdh2*, *Elf5* in the neural differentiation and *Pax6*, *Bmp4* and *Sfrp2* in EB differentiation).

The difference in phenotype between the *Mbd3Δ* and *Mta123Δ* ES cells did come as something of a surprise. In hindsight it really shouldn't have: we always knew that *Mbd3Δ* was a NuRD hypomorph, not a null. In contrast we believe the *Mta123Δ* cells to be complete NuRD nulls. If facilitating differentiation was the ONLY function of NuRD in ES cells, then yes, one would expect a more pronounced defect in the exit from pluripotency in *Mta123Δ* cells, and we probably wouldn't have learned much new about NuRD function. This unexpected phenotype has allowed us to identify a new function for NuRD in ES cells of reducing transcriptional noise and thereby facilitating fidelity of lineage progression, which had not been detected in the NuRD hypomorph (*Mbd3Δ*) ES cells.

6. The in vivo differentiation assay (ICM incorporation) of *Mta1/2/3* KO ESC is difficult to interpret. I encourage the authors to perform teratoma analysis of *Mta1/2/3* KO ESCs and compare results to *Mbd3* KO ESCs and WT ESCs as an unbiased, spontaneous in vivo differentiation assay.

We agree with this reviewer that a spontaneous differentiation assay is an important addition here. However a teratoma will report on the ability of a cell to differentiate in what is a completely inappropriate environment: ES cells are derived from peri-implantation stage embryos, but teratomas are formed on somatic tissues of adult animals. We know *Mbd3Δ* ES cells can differentiate in teratomas, which is consistent with their ability to respond and differentiate upon addition of retinoic acid (Kaji et al., 2006) (which again is not something normally seen by a pluripotent cell). We felt it more appropriate to conduct directed differentiation assays towards neuroectoderm (Figs 4 and 5) and mesoderm (Fig EV5) so that we can have a more focussed interpretation of the result, and to place them into a stage-appropriate setting, i.e. the early mouse embryo, to assess differentiation potential.

Teratoma assays require the use of live mice, and I would struggle to justify the use of more animals to the UK Home Office for this rather messy assay when other, more directed assays are available, and when the funding for this project has long since ended and I do not have animal funding I can justify using for this experiment. To provide an alternative unbiased spontaneous differentiation assay as requested by this Reviewer we have allowed control, *Mbd3Δ* and *Mta123Δ* ES cells to form embryoid bodies and monitored gene expression as these EBs were allowed to attach and differentiate in minimal media with serum (Figure 5C, D). Under these conditions ES cells can form a wide range of different tissues (Doetschman et al., 1985). This experiment shows that while mutant cells can downregulate pluripotency genes, they fail to properly activate markers of differentiated lineages such as ectoderm (e.g. *Pax6*), mesoderm (e.g. *T*, *Bmp4*) or endoderm (*Sox17*, *Gata6*).

Minor point:

The color key in Fig. 5A is hard to see and should be changed

We have changed the colour scheme in an attempt to make it more clear.

Referee #2:

Points of criticisms:

Figure 2:

While I agree with the conclusion that Mta1-3 bind largely in a similar manner the authors present pie charts that are potentially misleading as they give the false impression of distinct targets. Notably peaks that appear unique to one of the three proteins appear to have weak signal. A better comparison of the ChIPseq data is to provide pairwise scatterplots. These will likely reveal that "unique" peaks are mostly sites where the signal is barely below or above thresholds that is set by the peak recognition algorithm. Really unique sites should have strong signal in one protein and weak to no in another. Otherwise these can simply reflect different chip efficiencies.

The heat maps (Fig. 2D, E; EV2B) show that there is very little enrichment for other MTA proteins at "MTA only" peaks. Nonetheless both Reviewer 2 and Reviewer 3 suggested displaying the data as scatterplots which, we agree, provides much more information than a Venn diagram. New Figure 2A shows the three pairwise scatterplots for the MTA proteins. Thank you for making this suggestion.

Referee #3:

Burgold et al EMBO J

This manuscript explores the genetics and biochemistry of the MTA subunits of the NuRD complex. The authors perform biochemical and molecular experiments in ES cells to complement developmental experiments in embryos.

In general, the manuscript addresses a topic of interest to the chromatin community. The principal conclusion of the work is that MTA proteins act in a redundant fashion in ES cells to fine tune gene expression for development.

I find aspects of the biochemistry experiments to be incompletely described (see below) and other portions to be of poor quality.

The gene expression experiments focus on triple KO cell lines without directly testing the central conclusion of the manuscript - that the MTA proteins are redundant in terms of gene regulation. It seems to me a direct test would be to assess the transcriptional program in the single KO ES cells.

We did not intend to give the impression that the MTA proteins are redundant 'in terms of gene expression', but rather we demonstrate that they are functionally redundant in ES cells. This conclusion is based upon our findings that differentiation defects seen in triple

null ES cells can be rescued with any of the individual MTA proteins, and single- or double-mutant ES cells do not show the differentiation phenotypes.

We now include the analysis of our single mutant RNA-seq datasets in Fig. EV3. Different RNAseq experiments will always show some variability and while we can detect changes in expression in the Mta2-null and Mta3-null ES cells relative to their parent lines (Fig. EV3F,G), these differences do not prevent them clustering together with other wild type lines on a PCA plot. Specifically, whereas *Mbd3Δ* and *Mta123Δ* cells are distinct (Fig. EV3H). Given that single MTA mutant mice and some of the double mutants are viable, we concluded that any gene expression changes observed in single mutants are unlikely to be important for development. Rather than chase possible transcriptional differences with no apparent functional consequence we felt it was more important to focus on a mutant that showed a phenotype in our experimental system, specifically the triple mutant cells.

We have changed relevant text to ensure we state that the proteins are functionally redundant without giving the false impression that they are completely redundant for gene expression.

From a high level perspective, it would be useful to the readers of EMBO J for the authors to discuss why evolution has resulted in three MTA paralogs in vertebrates of their function(s) are redundant.

Again, the proteins have redundant functions in ES cells and during early development. Previous work from other labs has very clearly shown that the proteins do exert isoform-specific functions in mouse B-cells and human breast cancer cell lines. While we are certainly in no position to answer the question of why this protein family underwent amplification at the invertebrate/vertebrate boundary, we have included a brief discussion of this topic in the Discussion.

Specific comments to assist the authors in improving their manuscript:

Line 138 Each MTA protein was able to immunoprecipitated both of the other MTA proteins in addition to unmodified forms of itself.

The data does not support this conclusion. I see no evidence in 1B that MTA1 can IP MTA3.

We have replaced this figure with a much cleaner set of IP western blots. Mta3 is always difficult to detect in Mta1 IPs, whereas Mta1 is easily detected in Mta3 IPs. Mta3 is readily detected in Mta1 IPs by mass spectrometry (Fig 1C). This could be due to poor quality of the Mta3 antibody, but could also be because Mta3 is not terribly abundant in ES cell nuclei. We now mention this in the relevant part of the results section, and state “This could indicate that any Mta1-Mta3 containing NuRD complexes represent a relatively small proportion of nuclear Mta1, but a relatively large proportion of nuclear Mta3.”

- Please provide a method for nuclear extract production for this experiment. Are the co-IPs sensitive to nuclease treatment?

The IPs used for this new figure were repeated using benzonase, and the Methods section has been updated. Adding benzonase made no difference to the co-IPs.

- Data for MTA3 IP with MTA2-GFP is of poor quality. Please use a light chain specific secondary antibody (likewise coIP of HDAC2 and Rbbp4). The current data are not acceptable.

The figure has been completely replaced. We tried the light chain-specific secondary, but this actually made things worse. We ended up using a llama anti-GFP, which provided much more clear results.

- It would help the reader to interpret this data if the authors could indicate what percentage of input material is run for each IP in the Figure Legend.

We now state in the figure the percentage of input loaded.

IP Mass Spec

The methodology for the IP mass spec is also not described. Reference to a previous publication is not sufficient. Please describe the experiment

We have added this in the Methods.

- Are the IP's sensitive to nuclease treatment? Ethidium bromide in the IP (as described at PRIDE) perturbs B form DNA helical structure but in no way changes the fact that DNA is a polymeric anion capable of bridging otherwise discrete particles.

As the reviewer correctly states, we add ethidium bromide to our purifications, which serves to inhibit indirect, nucleic acids mediated protein-protein interactions during affinity purification. In addition, our affinity purifications are washed with 300 mM salt, which also efficiently disrupts protein-DNA interactions. We have previously published a methods paper in which we describe the effect of ethidium bromide for nuclear protein-protein interaction studies (Baymaz et al., 2014). Thus, all of the interactions we describe and present in our paper are direct protein-protein interactions that are not sensitive to nuclease treatment.

- I do not understand the volcano plot and its description on the PRIDE site. Threshold adjustment so that control values are not significant. This is unclear. I generally believe p value is related to hypothesis testing. How do control values test a hypothesis and get a p value? Please explain how this works. Are the p values adjusted for multiple testing?

We apologize for causing any confusion. For interactor identification, t-test-based statistics is applied on label free quantified (LFQ) mass spec data as described earlier (Hubner et al., 2010). First, the logarithm (log 2) of the LFQ values are taken, resulting in a Gaussian distribution of the data. This allows imputation of missing values by normal distribution, assuming these proteins are close to the detection limit. Statistical outliers for the Flag pull-downs in tagged mouse ESC and WT ESCs as control are then determined using two-tailed t-

test. Multiple testing correction is applied by using a permutation-based false discovery rate (FDR) method in the software we use to analyse our data, Perseus (Tyanova et al., 2016).

- I am unclear on the n for each experiment as reported in the evidence table on the PRIDE site. For example, there appear to be 6 separate experiments for MTA2-GFP mass spec, with 3 experiments for MTA2-GFP in MBD3 KO. There appear to be 3 experiments for MTA1 (all three conditions) and 6 for MTA3 (all conditions). I am unclear on why this is the case. Can one legitimately compare across analyses performed on different numbers of experiments? Doesn't this impact hypothesis testing? Does this signify that IP efficiency is vastly different with the different antibodies (FLAG v GFP)? Please clarify.

Again, we apologize for causing any confusion here. All the figures in the paper were generated by plotting 3 specific and 3 control pull-downs against each other. However, for some of these experiments we had more than 3 replicates and we decided to upload all these raw files to PRIDE. To avoid any confusion, we repeated the raw data upload to pride and only included the replicates, which we used to generate the figures shown in the paper.

- What is the difference in the evidence table between MTA2_nonGFP and MTA2_WT in the evidence table on the PRIDE site? By my very simple analysis of the evidence table at PRIDE, it appears that MTA1 IP brings down much more MTA1 than MTA2 (572 peptides v 414) and substantially more MTA1 than MTA3 (572 peptides v 192). One can make similar observations for the other pulldowns - leading to the likelihood that MTA proteins may be more likely to IP themselves than their paralogs. Please provide analysis to address this - the data are present here and the analysis would be of interest to readers of EMBO J.

Protein abundance cannot be directly derived from counting peptides. In our manuscript however, we make use of the iBAQ algorithm, which allows relative quantification of proteins against each in affinity purifications (methodology discussed in detail here: (Smits et al., 2013)). The reviewer raises an important point here, and it is correct that in AP-MS proteomics studies it is quite common to IP more of the bait protein compared to its interaction partners, some of these interaction partners may be washed away after the purification when using relatively harsh washing conditions and large (ml) washing volumes. The referee, however, is right when stating that MTA3 indeed seems to be less abundant in ESCs compared to the other MTAs. MTA3 seems to be particularly abundant in immune cells (Fujita et al., 2004).

- The identification of CHD4 in the mass spec is somewhat surprising - identification of CHD4 is really low. GATAD2A/B are recovered with something like expected frequency. Do the authors believe that the nearly 10-fold difference in abundance of a putative positive control is indicative of a meaningful experiment? Please explain. Readers of EMBO J would benefit from discussion of this point.

Based on our work and that of others (Bode et al., 2016, Low et al., 2016, Zhang et al., 2016), it has become clear that CHD4 is a peripheral subunit of the NuRD complex. It is thus conceivable that some CHD4 is removed from the core (HDAC-MTA-RbAp) during affinity purifications and subsequent stringent washes. Furthermore, some of the MTA proteins

assemble in complexes that do not contain CHD4 (Link et al., 2018, Zhang et al., 2018). Altogether, this results in relatively low stoichiometry values for CHD4 in MTA purifications.

Likewise, the structural data from the Schwabe group would predict a 1 to 1 molar ratio of MTA protein with HDAC. It seems more like 2 to 1 for MTA1 and MTA2. Please explain for the reader (and for this reviewer - I do not understand).

This issue is again most likely related to the fact that we tag the MTA proteins and we are thus pulling down MTA proteins more efficiently than its interaction partners. We now mention this in the legend to Figure 1D.

ChIPseq

- I am unable to assess the quality of this data without access to genome browser compatible data. The authors need to make this data available for review.

Data are now available at GEO: [GSE122833](https://www.ncbi.nlm.nih.gov/geo/query/acc.cgi?acc=GSE122833)

- The ChIP data have very different peak numbers for MTA3 as compared to MTA1 and MTA2 (approximately 1/2 the number) - is this an outcome of lower expression or different chip quality (as I do not have access to the chip data, I am unable to discern quality...)

We now include sequence quality data in Supplemental Table 2. There is no large difference in ChIP quality between the different MTA proteins, so it is most likely that the difference in peak numbers is due to the lower amounts of Mta3 in ES cells.

ES cell nuclei contain about half as much Mta3 as Mta1, and considerably more Mta2 (see plot below generated by measuring total protein abundance in ES cells) (Zhang et al., 2017).

- It would be informative to readers to see the distribution relative to Refseq Transcription start sites of the MTA1/2/3 peaks and the MTA1, MTA2, MTA3 only peaks.

This is now included as Figure EV2C. This plot shows that NuRD peaks occur at both promoter proximal sites as well as distal sites, while the vast majority of MTA-only peaks are found a distal sites.

- The authors should report the QC and filtering information for the ChIPseq. Number raw reads, passing filter, deduplication, uniquely mapped, ...

These data are now included in Supplemental Table 2.

- The reader would benefit from seeing the ChIP for the MTA proteins plotted as a scatter plot - MTA1 v MTA2, MTA1 v MTA3, MTA2 v MTA3. This type of plot permits the reader to not only see overlap, but also see magnitude of signal at overlapping regions.

This point was also raised by Reviewer 2 and we have now included these plots in Figure 2A. These provide much more information than the Venn Diagrams, and we thank the Reviewers for suggesting this.

MTA1,2,3 KO ES Cells

- Is MBD2 subject to the same type of protein-level control in the triple MTA KO ES cell line?

Yes, Mbd2 protein stability is also lost in *Mta123Δ* ES cells (Fig EV4A).

- It seems to me that GATAD2b abundance is greatly impacted in the MTA1,2 double KO. The bands in this blot are very confusing - heavy chain (?) is complicating the picture. Please guide the reader in the Figure legend to understand which band is the GATAD protein and which is antibody chain (?) in the IP. If, in fact, GATAD2B is lost in the MTA1,2 double KO, this merits discussion in the text. Is GATAD2A behaving similarly? What is the status of GATAD2A/B in the Mbd3 KO?

Figure 3C now includes further annotation to make the anti-Gatad2b panel easier to interpret. We have replaced Figure 3D with a new time course deletion blot including Gatad2a, Gatad2b, Mbd2 and Mbd3 to show that all four proteins are lost upon deletion of Mta3 (in an Mta1/2 double-null background). We also provide western blots of various mutant lines, including the *Mbd3Δ* cells, in Figures EV4A and B. Both Gatad2b and Mta3 are unstable in *Mbd3Δ* ES cells, Mta1 and 2 show partial stability as observed previously (Kaji et al., 2006, Reynolds et al., 2012) and Gatad2a appears unaffected. We now describe this in the Results section.

- It would be useful to see a gel filtration or sedimentation analysis of CHD4 in the MTA1,2,3 triple KO cell line to confirm and complement the coIP.

We agree this would be very useful to learn about the non-NuRD protein associations of Chd4. For the current study our aim was to show whether Chd4 could interact with components of the deacetylase subcomplex in the absence of MTA proteins, and the co-IP experiments clearly show that it does not. *Mta123Δ* ES cells will certainly be very useful in defining non-NuRD Chd4 activity in a subsequent study.

Gene Expression analysis

- Lines 251-255: this is a conclusion not supported by data. That a ratio of many genes is similar is not the same as no genes are impacted by the inclusion or absence of individual MTA subunits. This conclusion requires a different experiment. The current statement is true only at a global level in the absence of further experiments.

We are sorry for any confusion here. We did not say “No genes are impacted”. We meant that NuRD’s impact on transcription (globally is implied, but not stated) is not detectably altered by inclusion of different subunits. For example, the inclusion of Mta2 does not change the properties of NuRD such that it now activates more genes than it represses. In order to make our meaning clear, we have changed the final sentence of this paragraph to read as follows: “NuRD’s global impact on transcription is therefore not grossly altered by the inclusion or absence of individual MTA subunits.”

- The authors conclude that NuRD suppresses expression of lowly expressed genes. This analysis is based on fold change thresholds to define altered expression. This type of analysis is likely to bias towards low basal expression. Does analysis using a different metric (i.e. p value) lead to a similar conclusion?

The volcano plots in Figs. 4F and EV4D use both p-value and FC to define the significantly misexpressed genes. The programme used to identify differentially expressed genes, DESeq2, contains an inherent correction for lowly expressed genes for exactly the reason the Reviewer states:

<http://www.bioconductor.org/packages/devel/bioc/vignettes/DESeq2/inst/doc/DESeq2.html>.

The figures below show the expression data plotted for *Mta123Δ* ES cells and *Mbd3Δ* ES cells using two different methods of correction for lowly expressed genes. Genes are indicated as dots, red dots indicate significantly (adjusted p value of <0.05) misexpressed genes. The left panels use no correction, the second panels use standard correction from DESeq2 (as was used for figures in the paper: “normal”), the third panel uses the “ashr” method (<https://doi.org/10.1093/biostatistics/kxw041>) and the final panel uses “apeglm” (<https://doi.org/10.1093/bioinformatics/bty895>). Notably, *Mbd3Δ* ES cells also show predominant misexpression in the lowly expressed genes (e.g. mean counts $\leq 10^3$) though to a lesser extent than do *Mta123Δ* cells.

Differentiation trajectory

- It is unclear to me what the basis of the conclusion drawn in lines 356-357. It looks to me in Figure 5B as though the MTA KO cells follow the same developmental path as control cells, there are just fewer of them. Please elaborate for the reader on how the data shown support the conclusion drawn.

The mutant cells are able to maintain expression of Sox2 and show no evidence that they are adopting a trophoblast or primitive endoderm fate. They are unable to progress beyond this primitive state and keep up with the host cells as they are lost from developing embryos: we see no mutant cells surviving in the chimeras if they are allowed to develop beyond this stage of embryogenesis. Similarly, in the embryoid bodies mutant cells are unable to properly activate differentiation-specific gene expression programmes or produce the same wide range of cellular morphologies as do wild type cells, even though they are able to downregulate markers of pluripotency: we see no axonal projections or beating cardiomyocytes in mutant cultures even after 15 days (and counting) in culture. We now include these observations in the Results section to try to make this point clear.

Additional References:

Baymaz HI, Spruijt CG, Vermeulen M (2014) Identifying nuclear protein-protein interactions using GFP affinity purification and SILAC-based quantitative mass spectrometry. *Methods Mol Biol* 1188: 207-26

Bode D, Yu L, Tate P, Pardo M, Choudhary J (2016) Characterization of Two Distinct Nucleosome Remodeling and Deacetylase (NuRD) Complex Assemblies in Embryonic Stem Cells. *Mol Cell Proteomics* 15: 878-91

Bornelov S, Reynolds N, Xenophontos M, Gharbi S, Johnstone E, Floyd R, Ralser M, Signolet J, Loos R, Dietmann S, Bertone P, Hendrich B (2018) The Nucleosome Remodeling and

Deacetylation Complex Modulates Chromatin Structure at Sites of Active Transcription to Fine-Tune Gene Expression. *Mol Cell* 71: 56-72 e4

Boroviak T, Loos R, Bertone P, Smith A, Nichols J (2014) The ability of inner-cell-mass cells to self-renew as embryonic stem cells is acquired following epiblast specification. *Nat Cell Biol* 16: 516-28

Doetschman TC, Eistetter H, Katz M, Schmidt W, Kemler R (1985) The in vitro development of blastocyst-derived embryonic stem cell lines: formation of visceral yolk sac, blood islands and myocardium. *J Embryol Exp Morphol* 87: 27-45

Fujita N, Jaye DL, Geigerman C, Akyildiz A, Mooney MR, Boss JM, Wade PA (2004) MTA3 and the Mi-2/NuRD complex regulate cell fate during B lymphocyte differentiation. *Cell* 119: 75-86

Hubner NC, Bird AW, Cox J, Splettstoesser B, Bandilla P, Poser I, Hyman A, Mann M (2010) Quantitative proteomics combined with BAC TransgeneOmics reveals in vivo protein interactions. *J Cell Biol* 189: 739-54

Kaji K, Caballero IM, MacLeod R, Nichols J, Wilson VA, Hendrich B (2006) The NuRD component Mbd3 is required for pluripotency of embryonic stem cells. *Nat Cell Biol* 8: 285-92

Link S, Spitzer RMM, Sana M, Torrado M, Volker-Albert MC, Keilhauer EC, Burgold T, Punzeler S, Low JKK, Lindstrom I, Nist A, Regnard C, Stiewe T, Hendrich B, Imhof A, Mann M, Mackay JP, Bartkuhn M, Hake SB (2018) PWWP2A binds distinct chromatin moieties and interacts with an MTA1-specific core NuRD complex. *Nat Commun* 9: 4300

Low JK, Webb SR, Silva AP, Saathoff H, Ryan DP, Torrado M, Brofelth M, Parker BL, Shepherd NE, Mackay JP (2016) CHD4 Is a Peripheral Component of the Nucleosome Remodeling and Deacetylase Complex. *J Biol Chem* 291: 15853-66

Reynolds N, Salmon-Divon M, Dvinge H, Hynes-Allen A, Balasooriya G, Leaford D, Behrens A, Bertone P, Hendrich B (2012) NuRD-mediated deacetylation of H3K27 facilitates recruitment of Polycomb Repressive Complex 2 to direct gene repression. *Embo J* 31: 593-605

Smits AH, Jansen PW, Poser I, Hyman AA, Vermeulen M (2013) Stoichiometry of chromatin-associated protein complexes revealed by label-free quantitative mass spectrometry-based proteomics. *Nucleic Acids Res* 41: e28

Tyanova S, Temu T, Sinitcyn P, Carlson A, Hein MY, Geiger T, Mann M, Cox J (2016) The Perseus computational platform for comprehensive analysis of (prote)omics data. *Nat Methods* 13: 731-40

Zhang T, Wei G, Millard CJ, Fischer R, Konietzny R, Kessler BM, Schwabe JWR, Brockdorff N (2018) A variant NuRD complex containing PWWP2A/B excludes MBD2/3 to regulate transcription at active genes. *Nat Commun* 9: 3798

Zhang W, Aubert A, Gomez de Segura JM, Karupphasamy M, Basu S, Murthy AS, Diamante A, Drury TA, Balmer J, Cramard J, Watson AA, Lando D, Lee SF, Palayret M, Kloet SL, Smits AH, Deery MJ, Vermeulen M, Hendrich B, Klenerman D et al. (2016) The Nucleosome Remodeling and Deacetylase Complex NuRD Is Built from Preformed Catalytically Active Sub-modules. *J Mol Biol* 428: 2931-42

Zhang X, Smits AH, van Tilburg GB, Jansen PW, Makowski MM, Ovaa H, Vermeulen M (2017)
An Interaction Landscape of Ubiquitin Signaling. *Mol Cell* 65: 941-955 e8

Thank you for submitting the revised version of your manuscript. My apologies again for the delay in processing your revised manuscript as to protracted referee input. Your revised study has now been re-evaluated by the three original referees, please find their comments enclosed below.

As you will see, the referees #1 and #2 find that their concerns have been sufficiently addressed and they are now broadly favor of publication. Referee #3 states that the manuscript has been improved but remains more hesitant regarding data quality. Please note that to clarify this particular point we have also enquired back with referee #2, find his-her additional comments below. From this exchange we conclude that the remaining issue can be settled satisfactorily in a minor revision by additional data illustration and complementary discussion of the findings, introducing caveats where appropriate.

Overall, we are thus pleased to inform you that your manuscript has been accepted in principle for publication in The EMBO Journal, pending minor revision as outlined above.

REFEREE REPORTS:

Referee #1:

The authors have addressed all of my concerns and the revised manuscript is now suitable for publication.

Referee #2:

The revision fully addressed my concern in regards to data presentation.

Referee #3:

In revision, Burgold et al provide clarification of many points from previous review. The text modifications have greatly improved the manuscript.

The RNAseq data for individual MTA mutant ES cells is presented. I agree with the authors that these transcriptional profiles are very different from the triple mutant. It might aid the reader to have the major qualitative differences pointed out - for instance that MTA1 nulls have zero deg's, MTA2 KO have more downregulated genes than upregulated, ...

The ChIPseq data are less than impressive. Signal to noise ratios seem problematic, reproducibility across individual chip replicates seems somewhat less than ideal. I appreciate that ChIP of chromatin remodelers is challenging, but these data seem poor - particularly considering the use of epitope tags. I certainly do not agree with the authors assertion that MTA3 ChIPseq quality is similar to that of MTA1 and MTA2 - this is certainly not the case to my eye.

In the absence of better data quality, I am uncomfortable with some of the conclusions drawn regarding localization. Is this data necessary to the current story?

Referee #2, additional comments:

The data clearly support the main conclusion of differential and partly overlapping binding. Reviewer #3 is also correct in that chipping remodelers is very difficult, which needs to be taken into account when judging quality here. It is also important to note that data quality in this context only impacts the confidence that a particular site in the genome is bound but not the confidence that binding behaviour between two proteins is different at the scale of the genome, which is the main point.

I do not think that better quality is needed for this main conclusion and I would not know how to

achieve it. One way to accommodate the concern of the reviewer by the authors might be to acknowledge the limitation in the text and to add a supplementary figure that contains statistics about reproducibility between replicates. This would create the needed transparency for readers.

2nd Revision - authors' response

26th Mar 2019

We now indicate in the relevant part of the Results section that the Mta3 ChIP-seq gave lower coverage than did Mta1 or Mta2, most likely because it is expressed at lower levels, and that while we are confident in sites called at Mta3-peaks, we are less confident in declaring any site to be Mta3-free. We also include a correlation plot (Figure EV2) which shows that the Mta3 ChIP-seq is not an outlier as might be expected for poor quality data, and that replicate correlation is high, which will hopefully give the reader confidence in our ChIP-seq data.

Reviewer 3 asks that we comment on the single KO RNAseq data. We have considered this, but chose not to. We are wary about overinterpreting these data. If they do not cluster away from wild type cells on the PCA plot (Fig. EV4H), then we do not feel there is much to gain by describing various details such as the lack of misexpressed genes in the Mta1 KO, or how many go up or down in the others. Rather, we present the data for the readers to see these features for themselves and interpret as they wish, and of course the data are now freely available for anyone else to download and analyse.

The authors performed the additional requested editorial changes.

Corresponding Author Name: Hendrich
Journal Submitted to: EMBO Journal
Manuscript Number: EMBOJ-2018-100788